

# Ship plumes in the Baltic Sea Sulphur Emission Control Area: Chemical characterization and contribution to coastal aerosol concentrations

Stina Ausmeel[1], Axel Eriksson[2], Erik Ahlberg[1], Moa K. Sporre[1], Mårten Spanne[3], Adam Kristensson[1]

[1]Division of Nuclear Physics, Lund University, Box 118, 221 00 Lund, Sweden
[2]Ergonomics and Aerosol Technology, Lund University, Box 118, 221 00 Lund, Sweden
[3]Environment Department, City of Malmö, SE-208 50 Malmö

*Correspondence to*: Stina Ausmeel (stina.ausmeel@nuclear.lu.se)

**Abstract.** In coastal areas, there is increased concern about emissions from shipping activities and the associated impact on air quality. We have assessed the ship aerosol properties and the contribution to coastal particulate matter (PM) and nitrogen dioxide
($NO_2$) levels by measuring ship plumes in ambient conditions at a site in Southern Sweden, within a Sulphur Emission Control Area. Measurements took place during a summer and a winter campaign, 10 km downwind of a major shipping lane. Individual ships showed large variability in contribution to total particle mass, organics, sulphate, and $NO_2$. The average emission contribution of the ship fleet was 29±13 and 37±20 ng m$^{-3}$ to $PM_{0.5}$, 18±8 and 34±19 ng m$^{-3}$ to $PM_{0.15}$, and 1.21±0.57 and 1.11±0.61 µg m$^{-3}$ to $NO_2$, during winter and summer respectively. Sulphate and organics dominated the particle mass and most plumes contained undetectable
amounts of equivalent black carbon (eBC). The average eBC contribution was 3.5±1.7 ng m$^{-3}$ and the absorption Ångström exponent was close to 1. Simulated aging of the ship aerosols using an oxidation flow reactor showed that during a few occasions, there was an increase in sulphate and organic mass after photochemical processing of the plumes. However, most plumes did not produce measurable amounts of secondary PM upon simulated ageing.

## 1 Introduction

Air pollution from shipping is a global concern due to its climate and health effects (Oeder et al., 2015; Brandt et al., 2013; Corbett et al., 2007; Eyring et al., 2010; Lack et al., 2011). In many places, there is an increase in shipping activities as a result of increased international trade. Ship emissions are an increasingly important source of air pollution, especially in coastal areas and harbours (Corbett and Fischbeck, 1997; Eyring et al., 2005). Ship emission properties, such as particle number and mass concentration, particle size, and chemical composition, depend on a variety of parameters and ships make up a heterogeneous mix of emission
sources. Most particles emitted from ships are in the sub-micrometre range, typically with a diameter below 100 nm (Lack et al., 2009). Studies have shown a decrease in mean particle diameter when switching to a lower fuel sulphur content (Betha et al., 2017; Zetterdahl et al., 2016) as well as a decrease in emitted particulate matter (PM) (Lack et al., 2009; Diesch et al., 2013; Mueller et al., 2015; Buffaloe et al., 2014). The International Maritime Organization (IMO) have regulated fuel sulphur content in several steps in recent years, from 1.5% to 0.1% mass fraction between the years 2010 and 2015 in sulphur emission control areas (SECA). The
fuel sulphur limit is still 3.5% outside SECA but is planned to be restricted to 0.5% in 2020. Studies of chemical composition of ship PM have shown that the mass is typically dominated by organic matter, sulphate, and black carbon (Zetterdahl et al., 2016; Cappa et al., 2014; Lu et al., 2006; Anderson et al., 2015; Beecken et al., 2014). The relative distribution of these species depend partly on fuel sulphur content (Lack et al., 2009). There are several other parameters which affect emissions, such as fuel type, operation conditions, engine load, engine properties, and maintenance, which makes ship exhaust a heterogeneous aerosol for
different ships and geographical locations (Anderson et al., 2015; Brandt et al., 2013). Ship exhaust also contains elevated levels of nitrogen oxides ($NO_x$, including $NO_2$ and NO), sulphur dioxide ($SO_2$), carbon monoxide (CO), carbon dioxide ($CO_2$), and volatile organic compounds (VOCs). Concentrations of $CO_2$ can be used to estimate emission factors of gases and particles. $NO_x$ emissions from ships have also been shown to depend on, at least, both fuel type and on ship speed (Beecken et al., 2014; IPCC, 2013).



One way to characterize and quantify ship emissions is through ambient measurements in coastal areas, downwind of
shipping lanes. This method makes it possible to register an increase in aerosol levels and potential exposure in an area when
individual ship emission plumes pass the measurement station. Other methods include e.g. measurements on laboratory engine
emissions (Anderson et al., 2015; Kasper et al., 2007; Lyyränen et al., 1999; Petzold et al., 2010) or measurements on board or
following a sailing ship, intersecting the emission plume (Chen et al., 2005; Murphy et al., 2009; Petzold et al., 2008; Aliabadi et
al., 2016; Lack et al., 2011). However, while these methods can provide detailed knowledge on fresh emissions from a specific ship,
they do not give information about the variety of particle properties between different ships, how the plume evolves during transport
in the atmosphere, human exposure over land areas, and these methods can be more cost-intensive. By measuring in ambient
conditions on the coast, emissions from a large part of the shipping fleet can be captured, and atmospheric measurements are needed
to give information on emissions, dilution, and impact on environment and local air quality. Atmospheric measurements of elevated
$CO_2$ concentrations close to (less than a few minutes downwind) shipping lanes, can give emission factors during atmospheric
conditions, which differ from testbed conditions. It is difficult to simulate atmospheric dilution in testbed experiments, which has
large effects on nucleated nanoparticles.

To date, a number of atmospheric studies of individual ship plumes have been conducted in harbour areas (Alföldy et al.,
2013; Healy et al., 2009; Jonsson et al., 2011; Lu et al., 2006; Westerlund et al., 2015), and also in the Arctic (Aliabadi et al., 2015).
One study of aged plumes from a shipping lane has been performed by Kivekäs et al. (2014), outside the west coast of Denmark,
measuring plumes with an atmospheric age of about one hour. In the study by Kivekäs et al., ship plumes were measured at a coastal
station about 25-50 km from the shipping lane, where there was good potential to study the impact of ship emissions on land
concentrations and how particles are aged during semi-long range transport. However, the authors suggested shorter distances to be
able to detect elevated particle number concentrations from each individual ship passing along the lane when winds blew from the
ships to the coastal stations. In this study, we have performed measurements 10 km (corresponding to approximately 30 minutes of
plume aging) downwind of a major shipping lane in southern Sweden. With this setup, we were able to measure elevated particle
number concentration for a majority of the ship plumes. The distance is nevertheless large enough to represent typical shipping
lanes around the globe which influence inland air-pollution, as well as to observe some effects of particle aging.

The measurements presented here were performed in the Baltic Sea SECA during 2016 in order to study ship emission
properties after the newest regulation of fuel sulphur content (0.1% by mass). In a report by Mellqvist et al. (2017), the compliance
levels to the most recent SECA regulations was studied in the nearby region of where our measurements were conducted. During
the years 2015 and 2016, the compliance level was 92 %–94 % in the region around Denmark. The method for individual ship plume
identification and the contribution to particle number concentrations  have been described in detail in Ausmeel et al. (2019). In the
current paper however, we report the contribution from ships to local particle mass concentrations and chemical composition
(organics, sulphate, black carbon), and $NO_2$, as well as the effects of additional aging simulated with an oxidation flow reactor. The
results complement previous studies in two ways. Firstly, due to the new measurement location at an intermediate distance from the
shipping lane. Secondly, due to the measurements being performed after the recent fuel sulphur regulations within SECAs, which
was introduced on January 1, 2015. The estimation of how ship traffic along a major route contributes to the coastal particle
concentrations can contribute to the development of aerosol dynamic process models, regional aerosol particle models, health
assessment models and epidemiological studies.

## 2 Materials and methods

The field site and the measurement methods has been described in Ausmeel et al. (2019) and is only briefly outlined here. The
measurements took place at the Falsterbo peninsula, Southern Sweden, during January-March and May-July, 2016. This location is
within the Baltic Sea SECA. In order to measure particle number size distribution, and estimate mass concentrations for particles
with a mobility diameter up to 0.15 μm (PM$_{0.15}$) as well as 0.5 μm (PM$_{0.5}$), a custom built scanning mobility particle sizer (SMPS)



was used (DMA, Hauke type medium, custom built; CPC 3010, TSI Inc., USA) (Svenningsson et al., 2008). The scan time of the DMA was two minutes and the particle size range measured was 15-532 nm. SMPS size distributions were corrected for sampling losses using the Particle Loss Calculator tool (von der Weiden et al., 2009).

The particle chemical composition was evaluated online with a Soot Particle Aerosol Mass Spectrometer (SP-AMS, Aerodyne Research Inc.) (Onasch et al., 2012). The SP-AMS was alternately run in single and dual vaporizer mode during winter

and only in single vaporizer mode during summer. In single vaporizer mode, particles are flash vaporized upon impaction on a heated (600 °C) tungsten surface. In the dual vaporizer mode, particles containing refractory black carbon (rBC) are vaporized by an Nd:YAG laser (1064 nm). The vapours are then ionized (70 eV electron ionization) and detected in a high resolution time of flight (HR-ToF) mass spectrometer. The SP-AMS ionization efficiency with respect to nitrate was calibrated using 300 nm ammonium nitrate particles and for rBC calibration, regal black was used. The relative ionization efficiency for ammonium (RIE-

NH4) was 3.8 in winter and 4.0 in summer, and the collection efficiency (CE) was assumed to be 0.5 for both campaigns. SP was not used in the summer in order not to affect the detection of other compounds, since the BC levels were found to be low in the winter campaign. SP-AMS data analysis was performed using Igor Pro 6.37 (Wavemetrics, USA), Squirrel 1.57I, and PIKA 1.16I. In addition to the SP-AMS measurements, equivalent black carbon (eBC) content was measured with online, filter based optical absorption methods, using a seven wavelength Aethalometer (model AE33, Magee Scientific) (Drinovec et al., 2015) and a 637 nm

Multi Angle Absorption Photometer (MAAP, Thermo Fisher Scientific) (Müller et al., 2011), both with a sample time of one minute. For the gaseous compounds, $CO_2$ was measured at 1 Hz with a non-dispersive infrared gas analyser (LI840, LI-COR), $NO_x$ was measured every minute using a chemiluminescence monitor (CLD 700 AL, Eco Physics), and $SO_2$ was measured every minute using a UV fluorescent $SO_2$ analyser (S.A AF22M, Environnement).

A potential aerosol mass (PAM) oxidation flow reactor (OFR) (Kang et al., 2007; Lambe et al., 2011) was alternately

connected before the SP-AMS, SMPS, and Aethalometer to simulate atmospheric aging during parts of the summer campaign. The reactor is a 13.2 l aluminium cylinder with two UV lamps mounted inside. The reactor produces high levels of ozone and hydroxyl radicals (OH), and has been shown to produce similar yields and mass spectra to those from traditionally used smog chambers (Bruns et al., 2015; Lambe et al., 2015). The flow through the reactor was 5 litres per minute, which gives an average residence time of 160 s. The particle instruments sampled air from the radial centre of the reactor while ozone was sampled from a perforated

Teflon tube ring. The reactor and bypass sampling was controlled using an automated 3-way valve which was switching every 30 minutes. The first five minutes of ambient data and the first ten minutes of reactor data were not analysed, in order to give the sampling line time enough to stabilize flows. In some previous field studies with the PAM-OFR (Ortega et al., 2016; Palm et al., 2016), it has been placed outside with the end plate removed, directly sampling ambient air. This has been shown to minimize losses (Ortega et al., 2013). Due to the location, this was not possible in the present study. A fine metal mesh grid, which is usually installed

at the inlet side of the reactor to help develop a laminar flow, was removed in order to minimize losses. Further details about the simulated atmospheric aging, including calibration, particle losses, and fate of produced low-volatile species, are discussed in the S.I. (Fig. S3-5).

Individual ship plumes were extracted from the data set based on a set of criteria, which are described in the companion paper (Ausmeel et al., 2019). In brief, plumes were initially selected by inspection of the time series, choosing peaks in particle

number concentration where there was a clear increase above the background and noise level. This increase should not be longer than about 20 minutes and not shorter than about 5 minutes, to exclude other potential sources than ship plumes from the nearby lane. Plumes were only selected when the wind was blowing over the shipping lane to the measurement station. Further, automatic ship identification system (AIS) position data was used to confirm that the increase in particles was due to a passing ship. This was performed by calculating a trajectory of the emission plume from the ship using wind data. Background concentrations were

subtracted from the total plume concentration to get only ship emission contributions. The background levels in this analysis was defined as the baseline concentration from which the identified plumes can be distinguished. The background aerosol contains particles from several emission sources, including ship emissions from other regions which are well-mixed in the air. Hence, the





contribution from ships presented in this paper is not necessarily the total contribution from shipping, but only from one shipping lane of interest at a specific distance from the measurement site. For other variables, such as eBC measured with the Aethalometer,

aerosol mass measured with SP-AMS, and aerosol number concentration of larger particles (diameter above 530 nm) measured with APS, most individual plumes were not observable through a visual inspection of the time series. For these, the contribution during the plume duration was still calculated in the same way as for particle number concentration, followed by a statistical analysis (t-test) of whether this contribution was significantly larger than the background concentrations.

## 3 Results and discussion

### 3.1 Plume identification and general characteristics

An example of a time series from the SMPS (size distribution and number concentration) and SP-AMS (chemical speciation) during wintertime is shown in Fig.1. The middle panel also displays the times when ship plumes were expected to arrive at the measurement station based on ship position (AIS) and wind data. The plumes can visibly be distinguished from the background concentration in the time series of the particle number concentration (Fig. 1, upper and middle) as relatively short (ca 10 minutes) and intense peaks,

generally matching well with expected passages. Most particles are in the lower range of the size spectrum, below 100 nm in diameter (Fig. 1, upper). This is also reflected in the volume concentration (Fig. 1, middle), which does not increase during all plume events, since the small particles do not contribute largely to the total volume compared to the background concentrations. This means that most particles from ship emissions do not contribute directly to local particle mass concentrations while the contribution to number concentration is larger. Note that this is valid at the measurement site, ca 30 min downwind of the emission source.

Secondary aerosol formation can still contribute to larger mass further downwind, which is discussed in the Sect. 3.5 Simulation of atmospheric processing. In summary, several hundred ships passed the measurement station during the campaigns (approximately 70 per day), and those that were possible to attribute to a specific ship were studied in detail. Mass concentrations, $PM_{0.15}$ and $PM_{0.5}$, were calculated from particle number size distributions and evaluated for 113 plumes in the winter. Chemical content from SP-AMS measurements was evaluated for 15 and 18 plumes in winter and summer respectively. 76 plumes were evaluated for $NO_x$ and 100

plumes for eBC in the winter. The lack of eBC and $NO_x$ data as well size distributions in the summer was due to either instrument malfunction or unfavourable winds. Particles with an aerodynamic equivalent diameter larger than 530 nm did not contribute significantly to the particle number or mass concertation. This was shown by the aerodynamic particle sizer (APS 3321, TSI Inc. USA) measurements (0.5 – 20 µm). For all identified plumes, the contribution to $PM_{2.5}$ and $PM_{10}$ was assessed based on APS size distribution (assuming spherical particles with unit density). For the plumes observed in this study, there was no contribution to

$PM_{2.5}$ or $PM_{10}$ distinguishable from background levels.

A general feature observed in both measurement campaigns is that the particle properties varied from ship to ship. As described in Ausmeel et al. (2019), the ships evaluated in this study varied in size, load etc. The deadweight tonnage was 1-140 kilo tonnes, (10th and 90th percentile) with a median of 6 kt. The ship speed was 5-19 knots, (10th and 90th percentile) with a median of 11 kn. The ships observed in this study were mainly of the types cargo ship, tanker, and ro-ro ship. A correlation test with linear

least square regression was performed on all plume aerosol variables versus all ship properties. No significant correlation was found, even when normalizing plume concentrations for plume transport time. Hence, we could not conclude any relation between observed aerosol emissions 30 min downwind of the emission source and specific ship properties. However, for some variables there were few plumes observed, e.g. during summer when AIS data was lacking and for particle mass measurements (see table 1 and Fig. 2). A larger set of plumes would be needed for such relations to be observed, if present. In this study, there was no data available on

the specific fuel used by each ship. Another potential explanation for the variation in plume properties could be meteorological factors. In this study we have considered wind speed and precipitation, but no detailed analysis of the plume dispersion was performed and is outside the scope of this paper. Large variations between individual ships was also shown by Jonsson et al. (2011), at a measurement distance of 0.5-1 km from the ships.



The fact that Falsterbo is often not affected by a large particle volume (or mass) contribution from ships could potentially
be explained by the recently implemented SECA regulations, making ship owners improving or switching to other fuels. In this
study, we are only considering emissions from the shipping lane passing about 10 km from the measurement location. Depending
on wind direction, it is also likely that the background particle concentrations contain emissions from ships in other regions, making
the actual effect of ships on the local air quality larger. In a region with more ship traffic or in harbour areas, the local effects on
particle levels and consequently health implications will be of larger concern.

In Table 1, the average measured plume concentration is presented together with the background concentration (i.e. without
local plume contribution) as well as estimated corresponding daily and seasonal contributions of the shipping lane during winter
and summer to $PM_{0.15}$, $PM_{0.5}$, $NO_2$, eBC, and particle number (PN) levels. The average concentration for each plume was calculated
by integrating the total area under the plume peak. The values were then normalized by the plume duration to give an average plume
peak concentration. All ship passages that resulted in an elevated particle number concentration and which could be connected to
an individual ship with AIS were included in the calculation of the average contribution from the fleet. That is, even if no plume
peak was observed with e.g. BC measurements, it was still included in the calculation of the average BC. A t-test was performed to
check that the value of the BC contribution within the plume was significantly higher than the background. The daily and seasonal
values are based on AIS data, which showed an average of 73 and 63 ships passing per day in winter and summer respectively.
During periods when the wind blows from the Øresund Strait (i.e. across the shipping lane), the Falsterbo site is affected by the
nearest shipping lane approximately 51% of the time in the winter, and 44% in the summer, based on the average observed plume
duration of 10 min. Based on historical wind data from the last 20 years (SMHI), the wind intercepts the shipping lanes in Øresund
Strait about 70% of the time in both summer and winter, which was used to estimate the seasonal contribution from ships presented
in Table 1. For the daily and seasonal estimates, it was assumed that the average contributions in Table 1 are representative for all
plumes. For calculation of the uncertainty in the daily and seasonal contribution, the uncertainty in aerosol number concentration
was estimated to 30 %, the uncertainty in particle loss estimation was 30 %, the variation in ship traffic density was 17-34 %, and
the uncertainty in seasonal wind pattern was estimated to 5 %. These values were used to calculate the total uncertainty with error
propagation, i.e. added in quadrature. For a ship plume event, the average concentration in the plume during the entire duration of
the plume was used as the contribution value of that ship at Falsterbo.

        The measured ambient concentrations of $SO_2$ and $CO_2$ are also shown, although individual plumes were not distinguishable
from the background. $SO_2$ is typically of interest in ship emission studies, due to the fuel sulphur content, especially outside SECAs.
At the current distance of ca 10 km from the shipping lane, $SO_2$ plume concentrations were too diluted to be detected and separated
from ambient background levels in the winter campaign with an instrument noise level of 0.5 ppb. No measurements of $SO_2$ were
performed during the summer campaign. It is still possible that there is a ship contribution to $SO_2$, but the individual peaks are
smeared out so that it appears as a general increase in background level. $SO_2$ concentrations in Falsterbo were in general low, below
1 ppb. Ship contributions to $CO_2$ were also not detected, with an instrument noise level of less than 1 ppm. Therefore, it was not
possible to calculate emission factors at this distance from the shipping lane.

**3.2 Contribution to particle mass concentrations**

Particulate mass concentrations were calculated from the SMPS size distributions, assuming spherical particles with a density of 1.5
g cm$^{-3}$ (Matthias et al., 2010). Mass concentration values presented here are given in $PM_{0.15}$ or $PM_{0.5}$ (particulate matter in the in the
range 0-150 nm or 0-500 nm). The $PM_{0.15}$ contribution was included in this study to be able to compare with Kivekäs et al. (2014),
who also presented this contribution. Figure S1 shows the frequency distribution of the 113 individual contributions of plumes to
$PM_{0.5}$ during the winter campaign. Most ships have a small contribution to PM, of less than 100 ng m$^{-3}$. A few ships can be regarded
as relatively high mass contributors. It cannot be concluded from our results, whether the high contributors to observed plume peaks
at the coastline were non-compliant to the SECA regulations. Based on the number of plumes which resulted in high PM
concentrations (either BC, organics or sulphate, or all of these), compared to the number of plumes that should have intersected the



measurement site based on AIS and wind data, about 2-5 % were detected by our aerosol instrumentation. This number is in the same order of magnitude as the level of non-compliance reported by (Mellqvist et al., 2017), and could be related to the potentially higher emissions from such ships. In order to link these plumes to non-compliance, a more detailed study of the stack emission and the fuel properties (or use of exhaust scrubbers) is needed. Using the average PM for all ships, the contribution to the daily $PM_{0.15}$

and $PM_{0.5}$ levels in Falsterbo are around 8% and 1%, respectively. The contribution during the summer and winter measurements (assuming winds carrying ship emissions during 70% of the year) was ca 8-10% and 1% for $PM_{0.15}$ and $PM_{0.5}$ respectively (Table 1). Hence, we show that ships in this part of the Baltic SECA generally contribute with low PM concentrations. However, although mass contributions are low in this study, they can still be higher in areas with even more intense ship traffic, or close to harbours.

In a similar study to ours by Kivekäs et al. (2014), their reported $PM_{0.15}$ values were 100 ng m$^{-3}$ within plumes, and 23 ng

m$^{-3}$ daily contribution. This compares well to our values of about 50 ng m$^{-3}$ within plume and 26 ng m$^{-3}$ daily contribution. There is a plausible reason why the contribution to $PM_{0.15}$ should be similar in both the Kivekäs et al. (2014) and the current study. Namely, on the one hand, the ships were larger (had higher gross tonnage and deadweight) and the measurements were performed before the new 2015 sulphur content regulation in the SECA in the Kivekäs study and thereby likely had higher particle mass emission factors than at Falsterbo. On the other hand, the distance between ships and the station is larger in the Kivekäs study, suggesting that plumes

are more diluted. Hence, these effects are likely cancelling each other out, why the absolute plume $PM_{0.15}$ contribution becomes similar in both studies. According to Kivekäs et al. (2014), the maximum distance between ship and measurement site, for which plumes were still visible as an increase in number concentration, was about 50 km, and they suggested measurements to be performed at distances shorter than 45 minutes of transport time. In this study, we registered all plumes in particle number concentration at a distance of less than 45 minutes. However, mass concentrations and some gaseous compound were still not

detectable for all ships.

In a model study by Karl et al. (2019), three different regional chemistry transport model systems were used to study the influence of shipping in the Baltic Sea region, including the Øresund region in which our measurements took place. The maximum annual mean $PM_{2.5}$ contribution from shipping in the Øresund region was reported as 0.9 µg m$^{-3}$, corresponding to a relative contribution of 10 %. From our observations, ship plumes contributed with 0.029 ± 0.013 µg m$^{-3}$ (winter) and 0.037 ± 0.020 µg m$^{-3}$

(summer). Since we only observe the contribution from a single shipping lane, the comparatively low $PM_{2.5}$ contribution is not unexpected. However, the factor 20-30 difference between our observations and the contribution modelled by Karl et al. (2019) is still large. This could potentially be explained by the emission inventory used in model simulations being older than the most recent SECA regulation. Further studies of SECA regions and the typical fuels used within these would be valuable for confirming a potential decrease in the total PM contribution from ships due to the sulphur regulations.

**3.3 Contribution to NO₂**

Plumes contained about 0-10 µg m$^{-3}$ of NO₂, with a similar distribution among ships as for PM concentrations, i.e. that a few ships contributed with high concentrations while the majority of the plumes were diluted to below the detection limit of the instrument (0.1 ppm). The plume NO₂ concentrations, with background subtracted, are shown in Fig. S2, and average NO₂ values are presented in Table 1. There was in general no increase in NO concentration associated with the ships in this study. One instance of elevated

NO was recorded and attributed to a pleasure craft passing very close to the station (less than 1 km away and about 5 minutes of plume transport). This type of ship is not representative for the fleet in general, and not considered in further results or discussion. Except for this case, all contribution to $NO_x$ was in the form of NO₂ both during winter and summer, and during day and night-time. This implies that the plume transport in this case results in a well-mixed plume reaching the coast line. In a study by Karl et al. (2019), the ship-related NO₂ concentrations in the Baltic Sea region were evaluated using three regional chemistry transport model

systems. It was found that the contribution was 3-5 µg m$^{-3}$ along the main shipping routes. Our observed seasonal contribution in Falsterbo of 1.21 ± 0.57 µg m$^{-3}$ (winter) and 1.11 ± 0.61 µg m$^{-3}$ (summer) are lower than the one presented by Karl et al. Since our measurement campaign is limited to one shipping lane, we are likely treating diluted ship emissions from further away as background



emissions, and hence our calculated contribution is likely a lower estimate. The measurements presented here are also limited in time and do not cover a full year of observations.

### 3.4 Contribution to BC and chemical composition

The contribution from ships to measured eBC during winter was 12 % compared to background eBC levels (Table 1). For eBC, a very small number of ships are contributing. Most plumes show no eBC contribution at all while still increasing the particle number and mass concentration to some degree, indicating other major particle components. Within the ship plumes, the average eBC fraction correspond to approximately 2 % of the total $PM_{0.5}$ mass. The absorption Ångström exponent (AAE) was calculated for the plumes, using seven wavelengths of the Aethalometer, after background subtraction, and was on average ~1, which is typical for fresh BC (Kirchstetter et al., 2004; Sandradewi et al., 2008; Zotter et al., 2017).

Previous studies have shown an increase in light absorption at shorter wavelengths in plumes, indicating a significant fraction of brown carbon (BrC) (Yu et al., 2018; Corbin et al., 2018; Corbin et al., 2019). This was not seen in our study in the Baltic Sea SECA, which is in line with Corbin et al. (2018) who showed that burning of heavy fuel oil resulted in both BC and BrC, while marine gas oil or diesel fuel resulted in negligible BrC fractions and an AAE close to 1. Heavy fuel oil is not expected to be used as a fuel within SECA. For the MAAP, the plumes were not distinguishable from the background due to the detection limit of the instrument. The rBC content was also measured with the SP-AMS in the winter campaign, showing similar results with low contribution in general. There are however studies that have shown a significant BC fraction in ship exhaust. Lack et al. (2009) reported an average particle composition of 15 % BC, 46 % sulphate, and 39 % organic matter. Cappa et al. (2014) measured emission factors of BC from a ship running on low-sulphur marine gas oil and found extremely small sulphate fractions, while organic matter and BC dominated the particulate matter, with approximately 63 % BC at a speed of 6.9 kn, and 25 % at a speed of 12 kn. Hence, the average BC fraction of 2 % observed in Falsterbo appears to be relatively low, even with an average speed closer to 12 kn. It has been shown that BC emission factors depend on operating conditions, fuel quality and the potential use of scrubbers for reducing sulphur emissions (Lack and Corbett, 2012). Jonsson et al. (2011) showed that the non-volatile fraction of fresh ship emissions in a region close to Falsterbo (Gothenburg, west coast of Sweden) contributed to slightly less than 50% of the total particle population. However, it cannot be concluded whether the non-volatile fraction is consisting of soot particles or non-volatile organic compounds. In the model study of the Baltic region specifically (Karl et al., 2019), the shipping contribution to aerosol elemental carbon (EC) was found to be 0.03–0.04 µg m$^{-3}$ along the main shipping routes and in the main ports. This is a factor of ten larger than our measured contribution of 3.5 ng m$^{-3}$, similar to the discrepancy in $NO_x$, possibly due to the emission inventory being from before the most recent SECA regulation as well as the difference between the total influence from shipping contra that from one shipping lane.

Due to the small particle sizes and consequently the low mass concentration when diluted, the filter based measurement techniques are not optimal for investigating the presence of soot particles with our type of set-up (there could potentially be a high number concentration of small soot particles, while at the same time the BC concentration is low due to low light absorption). Also, the SP-AMS has a detection limit for particle size and hence some ship emitted particles at small sizes are not measured by the SP-AMS, and those that are, still contribute with a relatively small mass fraction. For further studies of ship BC emissions, other measurement techniques, such as the Single Particle Soot Photometer (SP2) which measures individual soot particles down to ~0.3 fg, could be useful. Also extracting microscopy pictures of the $PM_{0.1}$ fraction in fresh ship emission plumes could be useful for determining the presence of smaller sized soot particles.

The chemical composition of the ship plumes with relatively high PM contribution was provided by the SP-AMS measurements. In summary, the contribution to total PM is low with a very small or non-distinguishable refractory BC (rBC) fraction. Figure 2 shows the chemical composition of all plumes with a total mass concentration of more than 0.1 µg m$^{-3}$ (in order of decreasing mass). The average composition is inserted for each season. As seen in Fig. 2, when plumes have relatively high mass concentrations, the particles contain mainly sulphate and organics. For individual ships, there was a high variability in these



fractions, but there seems to be a higher fraction of organics in the summer based on the 33 observed plumes. There was also a large variation in total mass of the ship plume aerosol, where many of the plumes detected by the CPC were not seen at all in the volume and mass concentration time series (SMPS and SP-AMS), hence the relatively small number of plumes in Fig. 2. Most plumes were simply not above the SP-AMS detection limit (<50 nm) with the settings used, which was also confirmed with the SMPS particle volume. However, the particles with diameter below 50 nm will not contribute largely to the total mass and the results presented

here are hence focusing on the plumes with the largest PM contribution. Whether the difference in mass and specifically, the organic fraction, is significant between seasons require further measurements.

In one instance, the plume from an individual ship was detected multiple times. This is indicated with symbols in Fig. 2, where the ship passed the Falsterbo peninsula on three different occasions during the winter campaign. All plumes were detected during night time, and the ship-to-site distance and the transport time was similar (12-15 km and 20-30 minutes) during all occasions.

The chemical composition and the total mass is similar for the plumes from this ship, with similar fractions of sulphate and organics, compared to the large variability between different ships. This suggests that there is reproducibility in the method. For future studies, AIS data can be used to deduce which ship and engine types and fuels used are responsible for the increases in number and mass concentrations. For the organic fraction of the particles from all of the 33 ships plumes measureable by the SP-AMS, a larger amount of organics was seen in the plumes during summer. Comparing the mass spectra (Fig. 2) shows that the organic ship aerosol is very

similar in winter and summer, which strengthens the validity of the method used. The organic aerosol mass spectrum of the ship plumes in Falsterbo is similar to that measured in a laboratory study by Mueller et al. (2015), as well as in ambient conditions measured by Murphy et al. (2009) and Lu et al. (2006). The dominant hydrocarbon fragments are in general similar to those observed in aerosols from other traffic sources such as diesel emissions (Canagaratna et al., 2004). Elemental analysis of carbon, oxygen, and hydrogen content was performed according to Canagaratna et al. (2015) and resulted in similar O:C and H:C ratios in both

campaigns. The O:C ratio in the plume was 0.20 in winter and 0.21 in summer, while the H:C ratio was 1.89 in winter and 1.73 in summer. Hence, there is no indication of particularly stronger oxidation or aging of the plumes in the summer, possibly due to the relatively short transport time.

### 3.5 Simulation of atmospheric processing

Figure 3 shows an overview of the results from simulated aging with an OFR of ship plume aerosol during the summer campaign.

During the reactor period, the ambient aerosol particles were dominated by organics (78% by mass) and sulphate (14% by mass). The wind direction during the OFR experiment was mostly 90-180°, making the distance to ship lanes a few kilometres longer compared to periods with westerly winds. Also, the wind speed was typically lower during the OFR period, making the transport time of the plume about 90 minutes long compared to about 30 minutes during westerly winds. The reactor mainly changed the aerosol in two ways. A large number of small particles were produced and the O:C ratio was increased. The number of particles

increases because a super saturation of condensable vapours is produced as the aerosol enters the highly oxidizing environment. In the atmosphere, the production rate of these vapours is lower and instead of nucleating they would likely condense onto pre-existing particles. The ratio between reactor and ambient mass was mostly between 0.8 and 1.2. Much of this variation can be explained by the naturally changing ambient concentrations, since the enhancement is calculated as an average of reactor and bypass measurements varying with time. Notably there is no PM formation in the OFR in periods where no plumes are predicted. The

contribution of precursors from individual ships cannot be extracted from the data due to mixing in the OFR. Although there are periods of significant secondary PM formation (Fig. 3), considering the predicted number of plumes impacting the site, it is clear that most plumes did not contribute measurable amounts of secondary PM (as produced by the OFR). For example, none of the plumes (n=28) that were sampled during the last day (2016-06-07) of the experiment resulted in a net increase in $PM_1$ after simulated aging. However, for the period during which plumes produced secondary PM in the reactor (2016-06-03 and 2016-06-04), rather

high concentrations (several $\mu g/m^3$) were produced (see Fig. 3). Considering that none of the observed plume concentrations was



above one µg/m³ without OFR processing (see Fig. 2), this suggests that more PM forms in the plumes further downwind of the measurement site.

As shown in Fig. 3, some periods of secondary particle formation were observed in the reactor. The increase was due to sulphate and secondary organic aerosol (SOA) formation. Figure 4 shows one of these periods with a higher time resolution, where

several consecutive OFR engagements result in high (up to a factor ~2) increases in $PM_1$ from secondary aerosol formation. Particle volume increases during the periods with secondary PM formation in the OFR was a factor 1.5-2.5 and the absolute mass increase was several µg m⁻³. While both estimated volume from the SMPS and mass concentration as measured by the SP-AMS increased simultaneously during OFR processing, the magnitudes of the increase are not the same due to changes in the SP-AMS collection efficiency, as further discussed in the OFR section of the S.I. The increases in nitrate and ammonium (from oxidation of $NO_x$

followed by neutralization by ammonia) were moderate on an absolute scale. During the periods where no ship plumes were predicted, the difference between reactor and ambient measurements was close to zero, suggesting no net secondary formation from aging of the background air mass, and negligible particle losses on a mass basis.

Modelling of the fate of produced low-volatile species (Supporting Information and Fig. S4) suggests that a significant portion (~60-90%) of the oxidation products do not enter the particle phase due to the low condensation sink. There was no diurnal

trend in the enhancements, in contrast with previous OFR field campaigns in urban (Ortega et al., 2016) and forested areas (Palm et al., 2016). This may be caused by the air masses reaching the site already being somewhat aged and precursor concentrations being low. Further, the fact that the reactor was kept in an air-conditioned space increases the losses when it is colder outside (e.g. during nights), which was also seen by Ahlberg et al. (2019). No trend in the enhancement of particle mass with OH exposure was seen, likely due to the comparably low exposures. Ambient O:C and H:C ratios, commonly used as a proxy for atmospheric age, were on

average 0.62±0.15 and 1.53±0.15 (1 σ) respectively. The reactor produced an organic aerosol with a significantly higher O:C ratio (0.76±0.17) while H:C was not affected to any large degree (1.51±0.16). The O:C increase at times when aerosol mass was not increased (i.e., for the background aerosol and the majority of the plumes), suggests either heterogeneous oxidation or that SOA mass was formed and lost in similar magnitudes.

While the bulk of the secondary PM formed was due to organic compounds (see Fig. 3) two OFR engagements, shown in

Fig. 4, additionally resulted in high sulphate formation. The increase in sulphate upon processing, which was not observed for the vast majority of the approximately 100 plumes sampled, was possibly due to failure to comply with the Sulphur Emission Control Area regulations. No $SO_2$ data is available from the same period. OFRs can be used for qualitatively observing ships that contribute a lot to secondary aerosol inland, but should preferably be placed closer to the shipping lanes. Care must also be taken to ensure that reported secondary particle formation is not influenced by the background condensation sink (Palm et al., 2016; Ahlberg et al.,

360   2019).

## 4 Summary and conclusions

Ship emission plumes from the Øresund Strait were sampled with several on-line aerosol measurement techniques at a coastline in Southern Sweden during the winter and summer of 2016. A few up to a hundred plumes were analysed for particle mass contribution ($PM0_{0.15}$ and $PM0_{0.5}$), particle chemical composition, and gaseous $NO_2$. The aerosol particles were exposed to additional atmospheric

ageing using an oxidation flow reactor. The ageing of the background aerosol at Falsterbo in the oxidation flow reactor did not show significant increases in secondary mass, despite an increase in the O:C ratio. We suggest that the reason for this is that the background particles arriving at Falsterbo are already relatively aged. During the limited flow reactor measurements, there were a few cases with clear increase in sulphate and organic mass behind the flow reactor. However, the distance from ships during these days is relatively large and the contribution from individual ships is not clear, which means that further studies are needed to infer

how aged shipping particles can influence particle exposure. Falsterbo would be a good place to do further oxidative ageing experiments, but with slightly different setup than used in this study and for a longer time period. A common observation for all



aerosol parameters was large ship-to-ship variations in aerosol properties and plume concentrations, and these variations were not found to be correlated with any specific ship properties or plume transport time. Hence, the differences can be attributed to meteorological effects or variable exhaust properties. To determine which effect is most dominating, further studies are required.

For example, successful $CO_2$ measurements would make it possible to calculate emission factors of the aerosol species. To measure $CO_2$ in ship plumes, the measurements should either take place closer than 10 km from the emission source, or instruments with high sensitivity (better than < 1 ppm) must be used. During these measurement campaigns, ships in this part of the Baltic Sea SECA did generally contribute with low PM concentrations. The PM was dominated by organics and sulphate, in varying fractions, and the organic mass spectrum did strongly resemble that of typical diesel emissions. We were able to detect BC in plumes with one out

of three instruments measuring BC, namely the Aethalometer. The AAE of the ship BC vas close to 1, indicating diesel like fuel being common rather than heavy fuel oil, which is to be expected within a SECA. The contribution to eBC concentrations was relatively small, on average 3.5 ng m$^{-3}$, like the contribution to PM$_{0.5}$ exposure in general. The eBC fraction of the total mass was approximately 2 % in the diluted plumes, which is small compared to other ship emission studies. The BC contribution of ships sailing along shipping routes in this region is relatively small as has been demonstrated in this study. This indicates smaller health

impact due to BC compared to other anthropogenic BC sources. However, health effects due to other particle parameters might be significantly larger. For example, exposure to particle number concentration (Ausmeel et al., 2019), organic compounds, and NO$_X$ is higher. Although PM contributions are low in this study, they can still be higher in areas with more intense ship traffic, close to harbours, or in non-SECA regions. The regional impact of ships is higher than that measured in this study, since the background levels in Falsterbo also contain ship emissions from more distant shipping activities. This contribution is diffuse compared to the

individual ship plumes and therefore not possible to assess with our method. Hence, we expect an additional gaseous and primary and secondary aerosol particle contribution from ships sailing further away than about 30 km from the coastline. In addition, our OFR measurements indicate a possible secondary aerosol PM influence.

**Data availability**

The data sets used in this study are available upon request from the authors.

**Supplement**

Frequency distribution of the individual contributions of ship plumes. Technical details regarding the oxidation flow reactor, including measured and modeled losses. Comparison of SP-AMS and SMPS mass measurements during OFR periods.

**Author contributions**

SA analysed the aerosol data, was responsible for project administration during the summer campaign and prepared the manuscript.

AE was involved in the aerosol sampling and assisted in the data analysis and in the writing process. EA was involved in the aerosol sampling, conducted PAM-OFR experiments and analysis, prepared this section of the manuscript, and assisted in the data analysis and in the writing process. MKS generated trajectories used in the PAM-OFR analysis and assisted in the writing process. MS was involved in the aerosol sampling and in the writing process. AK designed the study, developed the model code, was responsible for project administration during the winter campaign and assisted in the writing process.

**Competing interests**

The authors declare that they have no conflict of interest.



**Acknowledgement**

This study was financed by the Swedish research council FORMAS (project no. 2014-951). The Crafoord Foundation (projects no.
20140955 and 20161026), is acknowledged for the contribution to the MAAP instrument funding and funding of the postdoc
position for the current studies. Fredrik Windmark, Swedish Meteorological and Hydrological Institute (SMHI), is acknowledged
for helping to provide AIS data. Paul Hansson, Henric Nilsson, and Susanna Gustafsson from the Environment Department at the
City of Malmö are acknowledged for helping preparing and setting up the measurements at Falsterbo. Dr. Kirsten Kling of DTU
and Dr. Antti Joonas Koivisto of NRCWE are acknowledged for helping with the summer campaign, and Fredrik Mattsson and
Anna Hansson, for helping with the winter campaign. Thank you also to Håkan Lindberg and the personnel from Falsterbo golf
court and Lennart Karlsson from Falsterbo birdwatching station who were willing to prepare a place for our measurement trailer,
and to the County Administrative Board of Skåne and Vellinge municipality for giving permission to measure in Flommen Nature
Reserve.

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





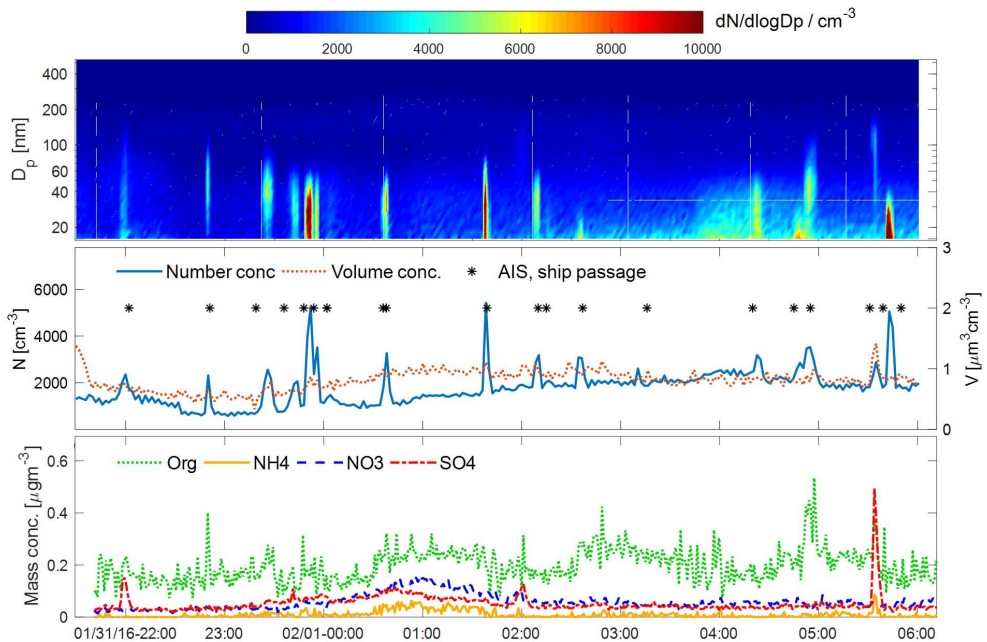

**Figure 1: Aerosol particle properties measured with an SMPS and an SP-AMS versus time (31 Jan. – 1 Feb. 2016), from measurements at the coastline in southern Sweden during an episode with westerly winds blowing from the Øresund Strait to the coastal station Falsterbo. The concentrations are those of the total aerosol, i.e. background concentrations are not subtracted. Top panel: 2D-Colour plot of total particle number size distribution from the SMPS. Middle panel: The total number- and volume concentrations from the SMPS and incidents of ship plume passages determined with AIS and meteorological data. Bottom panel: Concentration of the chemical constituents from the SP-AMS; total organics (Org), ammonium (NH₄), nitrate (NO₃), and sulphate (SO₄).**

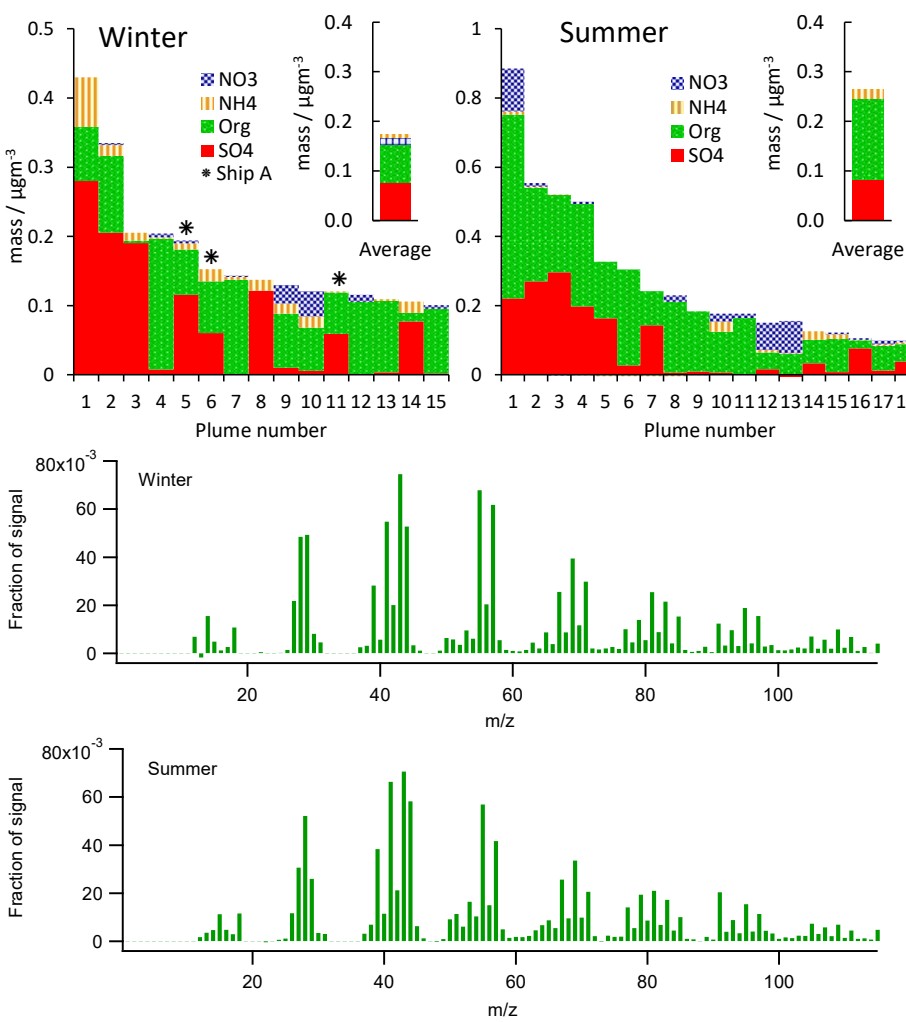

**Figure 2: Top panel: Mass concentration and composition of the particle constituents nitrate (NO₃), ammonium (NH₄), total organics (Org), and sulphate (SO₄) of ship plumes with an average mass content larger than 0.1 µg m⁻³, measured with the SP-AMS. Left plot shows plumes in winter and right shows plumes in summer sorted in decreasing order of total mass conc. Inserted bar plot shows the average of all plumes. The stars mark the plumes from the same ship passing at three different occasions (plume no. 5, 6, and 11). Middle and bottom panel: average organic mass spectra of plumes in the winter and summer measured with the SP-AMS.**





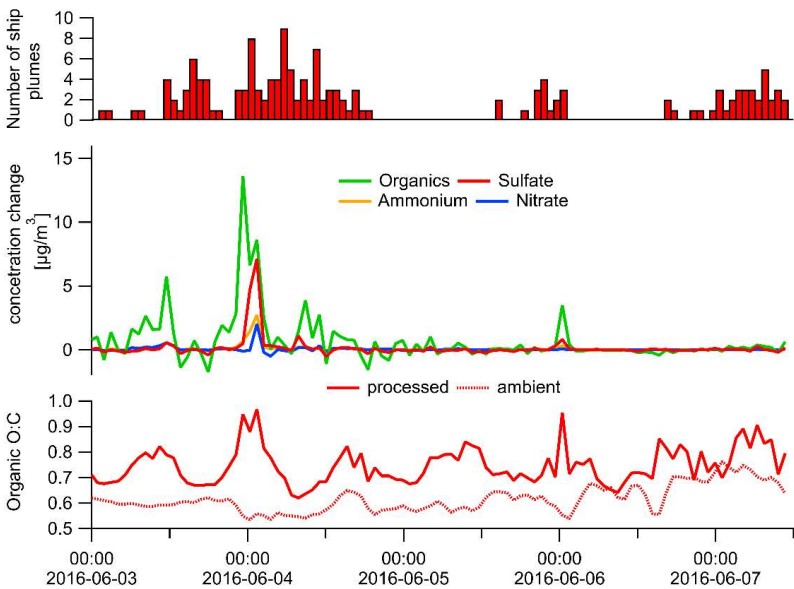

**Figure 3: Overview of an Oxidation Flow Reactor (OFR) experiment in the summer. Top panel: Number of ship plumes sampled. Middle panel: chemical species concentration change upon OFR processing. Bottom panel: organic O:C ratio after (processed) and before (ambient) OFR. Due to the alternating OFR/bypass sampling all data are hourly averages.**





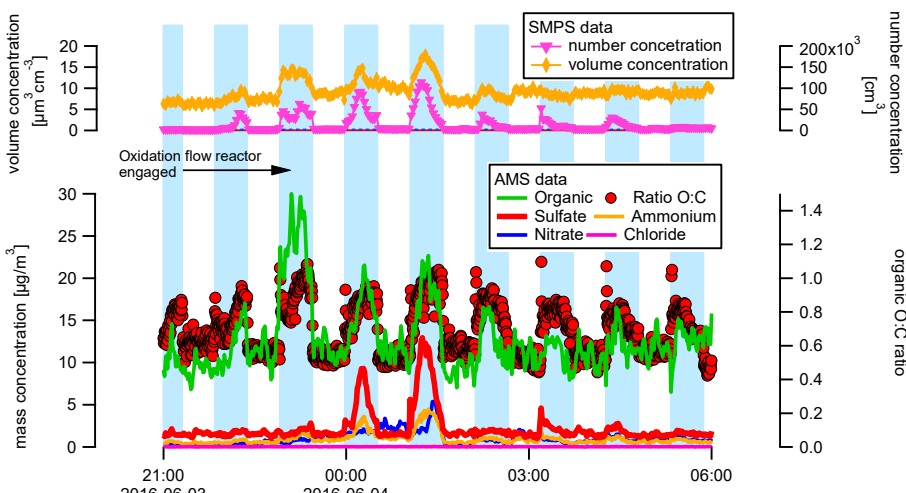

**Figure 4: Atmospheric processing simulated though an Oxidation Flow Reactor (OFR). OFR measurements are shown in light blue**
**background. Top panel: SMPS number and estimated volume concentration. Bottom panel: SP-AMS chemical species concentrations and organic O:C ratio.**



**Table 1: Average absolute contribution of particle mass (PM), NO₂, equivalent black carbon (eBC), particle number concentration (PN), SO₂, and CO₂ to local air quality due to ship plumes, from two measurement campaigns at the Falsterbo coastal site.**

| Season | Variable | Back-ground conc. [g] | Average plume conc. [g] | Contribution Daily [i] | Contribution Seasonal [i] | $n$ |
|---|---|---|---|---|---|---|
| Winter | $PM_{0.15}$ (ng m⁻³) [a] | 280 | 50 | 26 ± 12 | 18 ± 8 | 113 |
| | $PM_{0.5}$ (ng m⁻³) [a] | 2440 | 80 | 41 ± 19 | 29 ± 13 | 113 |
| | $NO_2$ (µg m⁻³) | 4.4 | 3.4 | 1.73 ± 0.81 | 1.21 ± 0.57 | 76 |
| | eBC (ng m⁻³) [b] | 210 | 9.9 | 5.0 ± 2.4 | 3.5 ± 1.7 | 100 |
| | N / cm⁻³ (CPC [c], 4 nm-10 µm)* | 1320 | 750 | 380 ± 180 | 270 ± 130 | 109 |
| | N / cm⁻³ (SMPS [d], 15-532 nm)* | 1200 | 700 | 360 ± 170 | 250 ± 120 | 113 |
| Summer | $PM_{0.15}$ (ng m⁻³) [a] | 500 | 110 | 48 ± 27 | 34 ± 19 | 8 |
| | $PM_{0.5}$ (ng m⁻³) [a] | 2720 | 120 | 53 ± 29 | 37 ± 20 | 8 |
| | $NO_2$ (µg m⁻³) | 3.6 | 3.6 | 1.58 ± 0.88 | 1.11 ± 0.61 | 17 |
| | N / cm⁻³ (CPC, 4 nm-10 µm)* | 2610 | 860 | 380 ± 210 | 260 ± 150 | 61 |
| | N / cm⁻³ (SMPS, 15-532 nm)* | 2530 | 1470 | 650 ± 360 | 450 ± 250 | 8 |
| Both | $SO_2$ (ppb) [e] | 0.20 | <DL [h] | - | - | - |
| | $CO_2$ (ppm) [f] | 430 | <DL | - | - | - |

* Value from (Ausmeel et al., 2019)

[a] Based on size distribution data.

[b] Based on Aethalometer data (880 nm).

[c] Condensation Particle Counter

[d] Scanning Mobility Particle Sizer

[e] Plume concentrations not distinguishable from background, instrument noise level is < 1ppm, according to manufacturer.

[f] Plume concentrations not distinguishable from background, instrument noise level is < 0.5 ppb, according to manufacturer.

[g] The background particle and gas concentrations (Background conc.) and the contribution due to ships (Average plume conc.) to different pollutants. Each value represents an average of a number of plumes ($n$) and are calculated from the ship plume peaks average concentration (i.e. concentration per unit time).

[h] Below detection limit (<DL).

[i] "Daily" values refer to days with wind directions where ships affect Falsterbo (mainly westerly) and "Seasonal" values refer to the average contribution observed at each campaign extrapolated over one season, including all wind directions.