# Peer review of "Ship plumes in the Baltic Sea Sulphur Emission Control Area: Chemical characterization and contribution to coastal aerosol concentrations"

_Atmospheric Chemistry and Physics, 2019_

## Referee Comment (RC1) · Matthias Karl (Referee) · 27 Dec 2019

The study evaluates the properties of ship-related aerosols and the contribution of a major shipping lane to the coastal particulate matter and nitrogen dioxide based on measurements in individual ship plumes during one summer and one winter season at a site in Southern Sweden. The method of ship plume identification is based on a previous publication of Ausmeel et al. (Atmos. Meas. Tech., 12, 4479–4493, 2019) and has been shown to give a robust estimate of the ship contribution to the ambient concentrations downwind of a shipping lane. In addition, the ageing of ship aerosols is investigated in an oxidation flow reactor (OFR), which occasionally showed an increase

in sulfate and organic matter upon photochemical processing in air associated with the ship plumes. The paper gives a brief overview of the results obtained in the field study and the OFR study, and is generally well written. The Introduction is very brief and does not give a comprehensive review of literature on the topic of measurements and chemical characterization of ship emissions, see my specific points (2+3) below.

The study has been performed in the context of the new sulphur regulation that requires fuel sulphur content equal to or less than 0.1% by mass in 2015. It should be made clearer in the discussion of the results how the paper complements previous measurements or modelling of the ship contribution to ambient air concentrations before/after the new sulphur regulation. The new regulation has led to changes in the type of fuel used or to installation of scrubbers: I think the Introduction should inform more about the changes of the ship operation (fuels, engine operation, scrubbers, etc.) and their expected impact on exhaust composition, in response to the new sulfur regulation. The paper should also discuss whether the finding of relatively low contribution from shipping to coastal particle phase concentrations might suggest that the taken measures have not only affected the sulphur components but also other constituents of the ship exhaust.

The distance of 10 km, which corresponds approximately to a plume ageing time of 30 min, seems long, even if it is considerable shorter than in a previous similar study by Kivekäs et al. (2014). At this distance, the particle number concentrations in the plume are already approaching the particle number concentrations in background air. Due to this distance, there is a chance that ship plumes from somewhat further away, like 20 km, are already further diluted and might contribute either to the background or to the plume signal of a "detected" ship. It should be explained how the method deals with other ships in the source corridor and the possibility of crossing or coinciding/indistinguishable ship plumes.

The method relies entirely on the accuracy and time resolution of the N measurements of the SMPS or particle counter. Kivekäs et al. (2014) apply a criterion for plume

detection that the excess N has to be larger than 500 cm-3 for detection of a ship plume (one hour plume ageing). Why is no such criterion used here? Meandering of the plume would also cause a fluctuation of N measured, which may lead to obscured plume detection. How did you deal with rapid changes of the background N?

The study does not discuss the impact of meteorological conditions and atmospheric stability on the plume detection. A higher and well-mixed boundary layer allows more vertical mixing of the plume and would lead to lower particle numbers in the plume that can be measured at the site. There are also some effects due to the location of the measurement site at the interface between the marine boundary layer and the boundary layer over land. Sensible heat flux associated with solar heating of the land surface can lead to the growth of a thermal internal boundary layer (ITBL) at the shoreline. Stable onshore wind flow advected over a cold water surface during the daytime, when passing the shoreline can be eroded by the ITBL, that can intercept an elevated ship plume and fumigate it rapidly to the ground (Lyons and Cole, 1973; van Dop et al., 1979; Hanna et al., 1984; Nazir et al., 2005).

Aromatic VOC might also be good tracers for the ship plume. Toluene levels measured by PTR-TOF have been used to detect contamination by ship exhaust in a ship-borne measurement campaign (Chang et al., 2011). Were VOC measured at the site or is it considered to complement future campaigns with PTR?

Murphy et al. (2009) reported a study with simultaneous shipboard and airborne measurements of the chemical composition and water-uptake of particulate ship emissions. One important finding of that study was that the in-plume organic-to-sulfate mass ratio did not change with increased plume ageing, indicating that the ship-originated particulate phase is not volatile enough to repartition back to the gas phase as the plume dilutes further. Please provide OC:SO4$_{dry}$ ratios measured in the ship plumes at Falsterbo (and compare to values in the literature) as they could be helpful in deriving emission factors of OC for the ship fleet in the Baltic Sea. The ratio could also give valuable information about the ships with high sulfate formation in the OFR, obviously

not complying with the new sulfur regulation.

**Specific Comments:**

1.) P. 1 line 18: Please give percentage fraction of plumes that did not result in measurable secondary PM.

2.) P. 2 line 48 - 50: It would be good to add a review of PM emission factors obtained by the $CO_2$ method, which is probably the most accurate method to infer real-world emissions, and compare to the PM emission factors obtained from testbed experiments.

3.) P. 2 line 50 - 51: Please explain: "It is difficult to simulate atmospheric dilution in testbed experiments, which has large effects on nucleated nanoparticles". Please explain the effects on nucleated nanoparticles in more details and add literature references. Does "atmospheric dilution" refer to the fast cooling and expansion of the exhaust plume at stack exit or the dilution due to atmospheric turbulence?

4.) P. 6 line 216 - 220: The explanation of the differences of PM0.15 in ship plumes to the study by Kivekäs et al. (2014) is somewhat speculative. What about the influence of different atmospheric chemistry regime or seasonal differences? The statement "these effects are likely cancelling each other out" is too strong. Simulations and observations of the particle number concentrations in ship plumes reported by Tian et al. (2014, figure 3 therein) show that after rapid dilution in the first minutes, total particle number concentration change only slowly between 30 and 60 minutes. This means that the effect of longer dilution period would be small. On the other hand, the ship contribution determined by Kivekäs et al. covers a larger source corridor, a fact that is not discussed here.

5.) P. 7 line 267 - 269: The average BC fraction of 2% of the total PM0.5 mass is very low compared to previous studies of ship exhaust, which may be explained in part by different fuels, operating conditions or the use of scrubbers. For a better understanding of this result, please provide the BC fraction, if only the plumes with detectable eBC

increase were included in the analysis.

6.) P. 9 line 339-340: Please discuss in more detail the (only) moderate increase of nitrate and ammonium in the OFR. Formation of ammonium nitrate requires the presence of sufficient ammonia, which may be the limiting factor during the time when the measurements were made. Thus, nitrate and ammonia could be higher during the spring season with more agricultural activity. The gas-phase/particle partitioning is also very sensitive to changes in temperature and relative humidity inside the reactor compared to ambient air.

7.) P. 9 line 351-353: How likely is it that heterogeneous oxidation of organic aerosols happens in the real plume ageing where the environment is much less oxidative?

**Technical Corrections:**

P. 4 line 143: "Chemical composition" would be better here.

Figure 1: Please add a panel with wind direction and wind speed below the current plots.

Figure 3: The orange line for ammonium is hardly visible.

Table 1: Please give the number of plumes in summer and winter somewhere in this table. It would be better to replace "average plume concentration" by "Delta plume" since the column gives the excess concentration due to the ship plume and not the in-plume concentration.

**References:**

Chang, R. Y.-W., Leck, C., Graus, M., Müller, M., Paatero, J., Burkhart, J. F., Stohl, A., Orr, L. H., Hayden, K., Li, S.-M., Hansel, A., Tjernström, M., Leaitch, W. R., and Abbatt, J. P. D.: Aerosol composition and sources in the central Arctic Ocean during ASCOS, Atmos. Chem. Phys., 11, 10619–10636, https://doi.org/10.5194/acp-11-10619-2011, 2011.

[Figure]

Hanna, S. R., Paine, R. J., and Schulman, L. L.: Overwater dispersion in coastal regions, Boundary-Layer Meteorol., 30(1-4), 389−411, https://doi.org/10.1007/BF00121963, 1984.

Lyons, W. A. and Cole, H. S.: Fumigation and plume trapping on the shores of Lake Michigan during stable onshore flow, J. Appl. Meteor., 12, 494−510, 1973.

Murphy, S. M., Agrawal, H., Sorooshian, A., Padró, L. T., Gates, H., Hersey, S., Welch, W., Jung, H., Miller, J., and Cocker III, D. R.: Comprehensive simultaneous shipboard and airborne characterization of exhaust from a modern container ship at sea, Environ. Sci. Technol., 43, 4626−4640, 2009.

Nazir, M., Khan, F. I., and Husain, T.: Revised estimates for continuous shoreline fumigation: a PDF approach, Journal of Hazardous Materials 118, 53−65, doi:10.1016/j.jhazmat.2004.10.008, 2005.

Tian, J., Riemer, N., West, M., Pfaffenberger, L., Schlager, H., and Petzold, A.: Modeling the evolution of aerosol particles in a ship plume using PartMC-MOSAIC, Atmos. Chem. Phys., 14, 5327–5347, https://doi.org/10.5194/acp-14-5327-2014, 2014.

van Dop, H., Steenkist, R., and Nieuwstadt, F. T. M.: Revised estimates for continuous shoreline fumigation, J. Appl. Meteorol. 18, 133–137, 1979.
* * *

---

## Referee Comment (RC2) · Anonymous Referee #2 · 12 Feb 2020

The paper aims to characterize and evaluate the contribution of ship plume to coastal aerosol by size distribution and SP-AMS measurements in a coastal site located in a recently classified sulphur emission controlled area (SECA). Besides, results on a simulated ageing of ship plume by an oxidation flow reaction are reported.

The paper present a large data set, but conclusions are not compelling. Here below some specific comments and suggestions to improve the discussion and especially to better constrain the conclusions.

The evaluated impact on of ship plume is really low, by reading the manuscript I do not understand what are the authors conclusions, is this due to the application of SECA

rules? Or to other meteorological effect? This aspect is fundamental to give guideline to environmental policies. Please authors improve the discussion on this aspect.

The result of the scarce increases of secondary aerosol obtained in the OFR measurements is not fully supported. I think that a comparison with data from other sites could help the discussion (e.g. Contini et al., 2011 and especially Perez et al., 2016 for the evaluation of secondary aerosol). Besides, the statement that background particles are already aged is not correct in my opinion, as the author state the time required to air masses from ship plume to the sampling site is 90 min in the measuring condition, this time is not sufficient for the ageing of aerosol, nether in summer. I think that information on the height of planetary boundary layer are fundamental to understand secondary aerosol formation processes and especially the real contribution of ship aerosol to background aerosol budget. By considering this aspect, conclusions can change substantially. I strongly suggest adding a discussion on the effect of PBL height on the contribution of both primary and secondary ship aerosol respect to background.

The paper can be published on ACP only if a deep discussion on the above reported points is added. Besides, few minor comments are reported here below.

Minor comments.

A map with the sampling site and surrounding areas with the urbanization level can be useful to interpret the data set.

Section 3.1 Plume identification and general characteristic seems more a methodology to recognize ship plume than results, I suggest moving this part in the Materials and methods section.

Lines 339-340. The sentence is not completely correct, it is true that nitrate arises from oxidation of NOx, but ammonium arises from neutralization of ammonia on both HNO3 and H2SO4, the latter is preferred over ammonium nitrate formation due to the lower

vapor pressure of sulfuric acid than nitric acid (Hauglustaine et al., 2014).

References

Contini D., A. Gambaro, F. Belosi, S. De Pieri, W.R.L. Cairns, A. Donateo, E. Zanotto, M. Citron. The direct influence of ship traffic on atmospheric PM2.5, PM10 and PAH in Venice. 2011. Journal of environmental management, 92, 2119-2129.

Hauglustaine D. A., Balkanski Y., and Schulz. M.: A global model simulation of present and future nitrate aerosols and their direct radiative forcing of climate. Atmos. Chem. Phys., 14, 11031–11063, 2014. https://doi.org/10.5194/acp-14-11031-2014.

Pérez N., J. Pey, C. Reche, J. Cortés, A. Alastuey, X. Querol. Impact of harbour emissions on ambient PM10and PM2.5 in Barcelona (Spain): Evidences of secondary aerosol formation within the urban area. 2016. Science of Total Environment, 571, 237-250.
* * *

---

## Author Response (AR1)

**Final author comments for manuscript acp-2019-1016**

We want to thank the editor for considering our manuscript for review and publication. We also thank both referees for their comprehensive and valuable comments to our manuscript. We think that the changes implemented after careful consideration of the comments have improved the content and clarity
5   of our manuscript.

This document contains the authors' response to the comments of both 'referee 1' and 'referee 2'. The comments are structures into three categories; 1) major/general, 2) minor/specific, and 3) technical, based on the specifications of the referees, or the judgement of the authors. All comments are followed first by a general author's response, and then by specifications about the changes done to the manuscript.
10  New text which is added to the manuscript is marked in *italic*, and all changes are shown in the new manuscript document with "track changes".

**Final author comment for Referee 1**

**1. Major/general comments:**

**Referee comment 1.1.**
15  **The Introduction is very brief and does not give a comprehensive review of literature on the topic of measurements and chemical characterization of ship emissions, see my specific points (2+3) below.**

**Author's response 1.1.**
We agree that the introduction could be extended slightly to include examples from the literature on
20  chemical characterization of ship plumes, and to guide the interested reader to further articles. We have added new text to the introduction, which gives examples from the literature of ship PM emission factors and chemical characterization. In order to make the introduction logical, some additional changes of the order of the paragraphs were made.

**Author's changes in manuscript 1.1.**
25  This is the new second paragraph in the section 1 Introduction:

"Ship emission properties, such as particle number and mass concentration, particle size, and chemical composition, depend on a variety of parameters and ships make up a heterogeneous mix of emission sources. Most particles emitted from ships are in the sub-micrometre range, typically with a diameter below 100 nm [*Lack et al.*, 2009]. Studies have shown a decrease in mean particle diameter
30  when switching to a lower fuel sulphur content [Betha et al., 2017; Zetterdahl et al., 2016] as well as a decrease in emitted particulate matter (PM) [*Lack et al.*, 2009; *Diesch et al.*, 2013; *Mueller et al.*, 2015; *Buffaloe et al.*, 2014]. *Studies of chemical composition of ship PM have shown that the mass is typically dominated by organic matter, sulphate, and black carbon. Zetterdahl et al. [2016] measured PM*

*emission factors on-board a ship in the Baltic Sea, running on heavy fuel oil (HFO) and low-sulphur residual marine fuel oil, respectively. The total and non-volatile PM emission factors were $0.17 \pm 0.03$ g (kg fuel)$^{-1}$ and $0.12 \pm 0.03$ g (kg fuel)$^{-1}$, respectively for HFO, and $0.06 \pm 0.03$ g (kg fuel)$^{-1}$ and $0.04 \pm 0.02$ g (kg fuel)$^{-1}$, respectively for the low-sulphur fuel. The black carbon emission factor ranged from 0.027 to 0.087 g (kg fuel)$^{-1}$, depending on engine load, and was slightly higher for HFO. Lack et al. [2009] reported emission factors for 43 ship plumes based on ambient measurements using aerosol mass spectrometry (AMS) for chemically resolved PM. They observed an average $PM_1$ emission factor of $3.32 \pm 4.04$ g (kg fuel)$^{-1}$, and specifically $1.21 \pm 1.50$ g (kg fuel)$^{-1}$ (36 %) was sulphate, $1.26 \pm 0.96$ g (kg fuel)$^{-1}$ (38 %) was organic matter, and $0.85 \pm 0.76$ g (kg fuel)$^{-1}$ (26 %) was black carbon. Sulphate and organic matter were linearly correlated with fuel sulphur content. In a study by Cappa et al. [2014], the ship plume PM was varying with ship speed, with an $EF_{PM1}$ ranging from 0.09 (slow speed, 2.9 knots) to 1.5 g (kg fuel)$^{-1}$ (high speed, 12 knots). For this ship, running on a low-sulphur marine gas oil, the PM sulphate content was below the detection limit of the AMS, while the primary organic matter (pOM) made up $53 \pm 14$ % of the total mass ($EF_{pOM}$ $0.39 \pm 0.44$ g (kg fuel)$^{-1}$), and BC made up $47 \pm 14$ %. Similarly, Shen and Li [2019] also found negligible sulphate emissions from marine diesel oil, which was dominated purely by organic and elemental carbon. In the same study, the use of HFO as fuel resulted in 75 % sulphate mass (including bound water), 21 % organic carbon, and the rest was elemental carbon and ash. Plume studies of 139 ships in a 1 % sulphur emission control area, presented in [Diesch et al., 2013], showed an average $PM_1$ emission factor of $2.4 \pm 1.8$ g (kg fuel)$^{-1}$, and specifically $0.54 \pm 0.46$ g (kg fuel)$^{-1}$ (23 %) was sulphate and $1.8 \pm 1.7$ g (kg fuel)$^{-1}$ (75 %) was organic matter. In a recent on-board study by Huang et al. [2018], it was also found that organic matter was the most abundant PM fraction (45-65 %), while sulphate content was low compared to the values listed above (2-15 %). The elemental carbon (or black carbon) PM mass fraction was low (1-6 %) for the main engine used for propulsion, while higher (20 %) for the auxiliary engine used to generate electricity. The PM composition, including other species than just sulphate, depend partly on fuel sulphur content [Lack and Corbett, 2012]. There are several other parameters which affect the absolute emission of PM as well as the particle composition, such as fuel type, operation conditions, engine load, engine properties, and maintenance.* Ship exhaust also contains elevated levels of nitrogen oxides ($NO_x$, including $NO_2$ and NO), sulphur dioxide ($SO_2$), carbon monoxide (CO), carbon dioxide ($CO_2$), and volatile organic compounds (VOCs) [Sinha et al., 2003; Chen et al., 2005; Alföldy et al., 2013; Moldanová et al., 2009; Huang et al., 2018; Cooper, 2001]."

**Referee comment 1.2.**

**The study has been performed in the context of the new sulphur regulation that requires fuel sulphur content equal to or less than 0.1% by mass in 2015. It should be made clearer in the discussion of the results how the paper complements previous measurements or modelling of the ship contribution to ambient air concentrations before/after the new sulphur regulation.**

**The new regulation has led to changes in the type of fuel used or to installation of scrubbers: I think the Introduction should inform more about the changes of the ship operation (fuels, engine operation, scrubbers, etc.) and their expected impact on exhaust composition, in response to the new sulfur regulation. The paper should also discuss whether the finding of relatively low contribution from shipping to coastal particle phase concentrations might suggest that the taken measures have not only affected the sulphur components but also other constituents of the ship exhaust.**

**Author's response 1.2.**

The specific explanation on how we complement previous studies is given at the end of the introduction:

80 "The results complement previous studies in two ways. Firstly, due to the new measurement location at an intermediate distance from the shipping lane. Secondly, due to the measurements being performed after the recent fuel sulphur regulations within SECAs, which was introduced on January 1, 2015. The estimation of how ship traffic along a major route contributes to the coastal particle concentrations can contribute to the development of aerosol dynamic process models, regional aerosol particle models, 85 health assessment models and epidemiological studies."

Regarding the specific changes in ship operation and fuel use in the Baltic region, we have added values from the literature, which were the most relevant we could find, on e.g. fuel use in the North and Baltic Sea during 2011, and information on scrubber use before and after the 2015 SECA. Additionally, our observed AAE of 1 is in line with emissions from distillate fuels, and the conclusion from this 90 observation has been made clearer as well.

Regarding general impact of fuel change, we have described this in in more detail in our new revised version of the introduction section (see Author's response 1.1. above). The discussion has also been extended to include more explanations of the observed low PM, including the effect of fuel sulphur reductions (see details in comment 1.1 by Referee 2).

95 ### Author's changes in manuscript 1.2.

We have revised the following parts in the manuscript section 3.1 Plume identification and general characteristics:

"Large variations between individual ships was also shown by Jonsson et al. [2011], at a measurement distance of 0.5-1 km from the ships. In this study, there was no data *currently* available on the specific 100 fuel used by each ship *or in the Baltic Sea in general. However, according to Jonson et al. [2015], the fuel distribution in the Baltic and North Sea in 2011 was around 74 % residual (e.g. HFO) and 26 % was distillate fuels (MDO, MGO). Further, Johansson and Jalkanen [2016] reported a 88 % decrease in $SO_x$ and 36 % decrease in $PM_{2.5}$ emissions from the year 2014 to 2015 in the Baltic Sea, based on AIS data and emission modelling. Strong decreases in $SO_x$ and $PM_{2.5}$ due to SECA implementations 105 have also been predicted and reported in e.g. [Kalli et al., 2013] and Jonson et al. [2015]. Regarding the use scrubbers to reduce airborne sulphur emissions, in the paper by Jonson et al. [2019] it is stated that there was an increase in the number of scrubbers used globally, from 77 vessels in 2014 to 155 in 2016. Out of these, 85 vessels with scrubbers were operating in the Baltic Sea area in 2016. This is a small fraction of the > 20 000 ships operating in the region [Johansson and Jalkanen, 2016].*

110 The fact that Falsterbo is often not affected by a large particle volume (or mass) contribution from ships could potentially be explained by the recently implemented SECA regulations, making ship owners improving or switching to other fuels. *The effect of sulphur regulations on the chemical composition is discussed further in Section 3.4.*"

"During the years 2015 and 2016, the compliance level was 92 %–94 % in the region around Denmark. 115 *Also Jonson et al. [2019] have shown that there is a strong indication that ships are complying, based on emission modelling before and after the 2015 SECA regulation.*"

We have added the following to the manuscript section 3.4. Contribution to BC and chemical composition:

"Previous studies have shown an increase in light absorption at shorter wavelengths in plumes, indicating a significant fraction of brown carbon (BrC) [Yu et al., 2018; Corbin et al., 2018; Corbin et al., 2019]. This was not seen in our study in the Baltic Sea SECA, which is in line with Corbin et al. [2018] who showed that burning of heavy fuel oil resulted in both BC and BrC, while marine gas oil or diesel fuel resulted in negligible BrC fractions and an AAE close to 1, *suggesting that distillate fuel is dominating in our sample*."

**Referee comment 1.3.**

**The distance of 10 km, which corresponds approximately to a plume ageing time of 30 min, seems long, even if it is considerable shorter than in a previous similar study by Kivekäs et al. (2014). At this distance, the particle number concentrations in the plume are already approaching the particle number concentrations in background air. Due to this distance, there is a chance that ship plumes from somewhat further away, like 20 km, are already further diluted and might contribute either to the background or to the plume signal of a "detected" ship. It should be explained how the method deals with other ships in the source corridor and the possibility of crossing or coinciding/indistinguishable ship plumes.**

**Author's response 1.3.**

We understand the referee's concern about plumes from longer distances not being visible at the measurement site. However, we found that almost all plumes (>95 %) were detectable with the method presented in the manuscript. We base this on the AIS data, which is obtained from the geographical area shown in the picture below (Fig. AC1).

[Figure]

Figure AC1. Location of the measurement station (circle with cross) at the Falsterbo peninsula together with ship traffic density (www.marinetraffic.com, 2016). Dashed square shows the area in which AIS positions are considered for ship identification. (From [*Ausmeel et al.*, 2019])

That is, we could identify a peak in particle number concentration (PN) for most ships passing this geographical area transmitting an AIS signal. This area includes the entire shipping lane described in the manuscript, including the fraction of ships sailing the furthest away from the coastline. In our previous paper [*Ausmeel et al.*, 2019], this method is validated. The Fig. AC2 below shows a typical time series of particle number concentration and identified ship plumes at the Falsterbo field site. This figure is representative for the entire measurement campaigns. We have added a sentence to the manuscript, describing the success rate of the method in order to avoid misunderstandings or similar concerns from future readers.

[Figure]

Figure AC2. Example of time series from total particle number concentration and the expected plume passages based on AIS and wind data (stars). (From [*Ausmeel et al.*, 2019])

It is still a valid and important comment that our study does indeed only deal with ships from the nearby region and not from much further away, e.g. from the Baltic or North Sea. We raise this point on several occasions; in the description of the method, in the discussion, and in the conclusions, specifically on p. 3-4, line 121-124, on p. 6, line 230, on p. 7, line 275, and on p. 10, line 388-390 in the first version of the manuscript. In order to make this unambiguous throughout the manuscript, we have clarified this also in the abstract and in the introduction.

Regarding crossing/coinciding plumes, it is described in the companion paper [*Ausmeel et al.*, 2019] that we ignore plumes that are close to each other in time (based on PN and/or AIS data) in order to keep the individual assessment of the plume contribution. Hence, this is not an issue for the interpretation of the results. However, this is not explicitly mentioned in the summary of the method in the current paper. We realize that this is a potentially important aspect for the interpretation of the results, especially in the ways described by the referee. We have made this clearer in the Materials and methods section.

**Author's changes in manuscript 1.3.**

The following two sentences have been changed in the manuscript abstract, and section 1 Introduction:

"The average emission contribution of the shipping *lane* was 29±13 and 37±20 ng m$^{-3}$ to PM$_{0.5}$, 18±8 and 34±19 ng m$^{-3}$ to PM$_{0.15}$, and 1.21±0.57 and 1.11±0.61 μg m$^{-3}$ to NO$_2$, during winter and summer respectively."

"In the current paper however, we report the contribution from a *major shipping lane* to local particle mass concentrations and chemical composition (organics, sulphate, black carbon), and NO$_2$, as well as the effects of additional aging simulated with an oxidation flow reactor."

The following two sentences have been added to the manuscript section 2 Materials and methods:

*"In order to assess individual ship plumes, no plumes that were overlapping or close to each other in time (less than five minutes) were included in the analysis."*

*"The expected plume passage based on AIS and wind data and the visible plumes in the particle counter*
180 *agreed excellently, and a large fraction (>95 %) of the plumes (including all plume events, not only the plumes finally used for analysis for which 100 % were confirmed with AIS) were detectable as a distinct increase compared to background particle concentrations."*

**Referee comment 1.4.**

185 **The method relies entirely on the accuracy and time resolution of the N measurements of the SMPS or particle counter. Kivekäs et al. (2014) apply a criterion for plume detection that the excess N has to be larger than 500 cm-3 for detection of a ship plume (one hour plume ageing). Why is no such criterion used here? Meandering of the plume would also cause a fluctuation of N measured, which may lead to obscured plume detection. How did you deal with rapid changes of**
190 **the background N?**

**Author's response 1.4.**

In Kivekäs et al. (2014), the minimum plume PN criterion is used to include all clear ship plumes and to exclude peaks due to other variability in the data. Due to the high time resolution (5 or 30 seconds) of the PN measurements in our measurements in Falsterbo compared to Kivekäs et al. (5 minutes), we
195 did not experience any issue with potentially including other variability in the data than ship plumes. The background was never changing rapidly compared to the duration of the ship plumes (average 10 minutes). Additionally, the availability of AIS data in our study made the confirmation of the ship plumes unambiguous.

With meandering of the plume, we assume that it refers to the plume potentially fluctuating in strength
200 or crossing the measurement site several times due to changing wind directions. However, this has not been seen at our site. Due to the geographical and meteorological conditions at the field site, we do also not expect such meandering transport of the ship plumes to any large extent during westerly winds which are associated with low pressure weather systems and higher wind speeds. These issues regarding accuracy and success of the PN and AIS data to detect and quantify plume contribution has already
205 been described in Author's response 1.3, and also in detail in the companion paper Ausmeel et al. (2019).

**Author's changes in manuscript 1.4.**

We have added the following parenthesis to the manuscript, section 2 Materials and methods:

The background level in this analysis was defined as the baseline concentration from which the
210 identified plumes can be distinguished *(the background concentration was always relatively stable during onshore wind, see Fig. 2, and hence the background subtraction was straightforward)*.

**The study does not discuss the impact of meteorological conditions and atmospheric stability on**
215 **the plume detection. A higher and well-mixed boundary layer allows more vertical mixing of the**
**plume and would lead to lower particle numbers in the plume that can be measured at the site.**
**There are also some effects due to the location of the measurement site at the interface between**
**the marine boundary layer and the boundary layer over land. Sensible heat flux associated with**
**solar heating of the land surface can lead to the growth of a thermal internal boundary layer**
220 **(ITBL) at the shoreline. Stable onshore wind flow advected over a cold water surface during the**
**daytime, when passing the shoreline can be eroded by the ITBL, that can intercept an elevated**
**ship plume and fumigate it rapidly to the ground (Lyons and Cole, 1973; van Dop et al., 1979;**
**Hanna et al., 1984; Nazir et al., 2005).**

**Author's response 1.5.**

225 The referee raises a relevant point regarding the impact of local meteorology when evaluating ambient
measurements. Indeed, the boundary layer height (BLH) affects the vertical mixing and consequently
the plume dilution. But since our aim is to quantify the contribution of ship plumes to the ambient air
pollution, the different meteorological conditions will simply represent different realistic scenarios in
the ambient air at Falsterbo. We are interested in presenting the absolute contribution at the coastal field
230 site, and do not attempt to say anything about the emission concentration at the ship plume stack. We
have highlighted this in the manuscript to avoid misunderstandings.

However, as the referee also seems to suggest, our results could still be affected if the dilution is so
strong that the plumes are not visible to our instrumentation during certain meteorological conditions.
After the referee's comment, we investigated the BLH at the field site during the measurement
235 campaigns in more detail, using BLH values from HYSPLIT data. The BLH varied between 70 and
1400 m (average including standard deviation was 570±290 m) in winter, between just a few and 1600
m (average including standard deviation was 490±320 m) in summer. The BLH was indeed typically
anti-correlated with aerosol concentrations on a longer time scale (hours to days). It could then be
expected that the plume concentration is affected to some extent by the BLH as well. However, the
240 BLH does not seem to affect our ability to detect plumes, but almost all (>95 %) expected plume
passages based on AIS data are observed by the particle counter, also during periods with high BLH
(see also Author's comment 1.3). Additionally, we investigated the particle number (PN) and
volume/mass (PM) concentration of the quantified plumes as a function of the boundary layer height,
see figure AC3. In this figure, the BHL does not appear to be correlated with plume concentrations, but
245 both high and low concentrations are observed during both high and low BLH. The result is inconclusive
and should need even more data to be able to draw any conclusions.

[Figure]

Figure AC3. Ship plume concentration (background subtracted) versus boundary layer height obtained from HYSPLIT. Left: Particle number concentration measured by CPC (circles). Right: calculated particle mass based on SMPS particle volume size distributions, assuming a particle density of 1.5 g cm$^{-3}$ (crosses). The lines represent least square linear fits to the data.

Regarding the effect of ITBL, we do not think this is an issue at this particular field site. In the references suggested by the referee, the geography is different from the case in our manuscript. The references all describe a situation which can occur at a rather long coast line, separating a large body of water and a large body of land. The measurement site in our manuscript is located at the tip of a peninsula, which is a relatively small land mass surrounded by water (see Figure AC1, or the map in the response to referee 2). Hence, we assess the boundary layer over the Baltic Strait and the shipping lane not to be very different from the boundary layer over the field site. Still, such meteorological effects are indeed something to consider when doing other coastal measurements. Therefore, we have added an additional description of the field site and commented on the potential effects of an ITBL in order to make this clearer. Additionally, a map of the nearby region has been added to the manuscript, further helping the reader to interpret the effects of the boundary layer over sea versus land (see Author's changes in manuscript 2.1, in response to referee 2).

**Author's changes in manuscript 1.5.**

The following text has been added to the manuscript section 2 Materials and methods:

*"When performing measurements along a coastline, the interface between atmospheric boundary layers over land and sea can cause a build-up of a thermal internal boundary layer at the shoreline. This has been described by e.g. Hanna et al. [1984] and is important to consider when doing plume measurements. However, due to the geographical surroundings at our field site, we do not expect this effect to be strong or exist at all, since the peninsula is a small land mass almost surrounded by water. The boundary layer height is likely a parameter which affects the dilution of the plumes and the background air and thereby the plume concentration. However, it seemed that boundary layer height did not affect the plume concentration in a systematic and reproducible way in Falsterbo as deduced using Hybrid Single Particle Lagrangian Integrated Trajectory Model (HYSPLIT) [Stein et al., 2016] with meteorological data from Global Data Assimilation System (GDAS). Since this study aims at describing the ship emission contribution to inland aerosol burden and not the concentrations at the exhaust stack, the effects of boundary layer height has not been pursued any further in this paper."*

**Referee comment 1.6.**

**Aromatic VOC might also be good tracers for the ship plume. Toluene levels measured by PTR-TOF have been used to detect contamination by ship exhaust in a ship-borne measurement campaign (Chang et al., 2011). Were VOC measured at the site or is it considered to complement future campaigns with PTR?**

**Author's response 1.6.**

The referee raises a good suggestion for future ship emission studies, but unfortunately our campaigns did not include VOC measurements.

**Author's changes in manuscript 1.6.**

The following sentence has been added to the manuscript, section 3.5 Simulation of atmospheric processing:

No $SO_2$ data is available from the same period. *In future ship emission campaigns, both $SO_2$ and VOC measurements would be useful for evaluation secondary PM formation and as tracers for ship emissions in general.*

**Referee comment 1.7.**

**Murphy et al. (2009) reported a study with simultaneous shipboard and airborne measurements of the chemical composition and water-uptake of particulate ship emissions. One important finding of that study was that the in-plume organic-to-sulfate mass ratio did not change with increased plume ageing, indicating that the ship-originated particulate phase is not volatile enough to repartition back to the gas phase as the plume dilutes further. Please provide OC:SO4dry ratios measured in the ship plumes at Falsterbo (and compare to values in the literature) as they could be helpful in deriving emission factors of OC for the ship fleet in the Baltic Sea. The ratio could also give valuable information about the ships with high sulfate formation in the OFR, obviously not complying with the new sulfur regulation.**

**Author's response 1.7.**

Out of the 33 plumes which were chemically resolved with AMS, there were 14 plumes with zero or close to zero ($< 0.01$ µg m$^{-3}$) sulphate content and two plumes with zero or close to zero organics. For the 17 plumes which contained both a sulphate and an organic fraction, the average OC:SO$_4$ ($\pm$ 1 stdv) ratio was 2.3 $\pm$ 3.2. We have added this value to the manuscript, together with a comparison with literature values.

**Author's changes in manuscript 1.7.**

The following text has been added to the manuscript section 3.4 Contribution to BC and chemical composition:

"For individual ships, there was a high variability in these fractions, but there seems to be a higher fraction of organics in the summer based on the 33 observed plumes. *Out of the 33 plumes which were chemically resolved with AMS, there were 14 plumes with zero or close to zero ($< 0.01$ µg m$^{-3}$) sulphate content and two plumes with zero or close to zero organics. The organic matter fraction (Org, OM) was translated to organic carbon (OC) using a conversion factor of 1.2, which is given by Canagaratna et al. [2015] for hydrocarbon-like organic aerosol (HOA). For the 17 plumes which contained both a*

*sulphate and an organic fraction, the average OC:SO₄ (± 1 stdv) ratio was 2.3 ± 3.2. Studies of single ships or test-bed engines show largely varying organics to sulphate ratios. In the study by Murphy et al. [2009], the OC:SO₄ ratio was 0.30 ± 0.01 in an airborne exhaust plume. The ship in that paper was*
320  *running on a 3 % m/m sulphur heavy fuel oil. In a study by Moldanová et al. [2009], OM:SO₄ was around 2.4 for HFO (or OC:SO₄ around 2.9, using the conversion factor OC = 1.2·OM given above), and in another study by Petzold et al. [2011], OM:SO₄ was around 0.38 for HFO and 16 for a marine gas oil (or OC:SO₄ around 0.46 and 19, respectively). Mueller et al. [2015] reported an OM:SO₄ which was around 3.8 for HFO and 515 for a marine diesel fuel (or OC:SO₄ around 4.6 and 618, respectively).*
325  *There are also studies reporting chemical composition for multiple ship, measured in ambient air, which is more comparable to our observations. Based on the average OC and SO₄ in 43 ship plumes measured by Lack et al. [2009], where the emission factors were 1.21 ± 1.50 g (kg fuel)⁻¹ (36 %) for sulphate, and 1.26 ± 0.96 g (kg fuel)⁻¹ for OM, the corresponding average OM:SO₄ is then 1.04 (or OC:SO₄ around 1.25). Similarly, for the 139 ship plumes observed by Diesch et al. [2013], the average*
330  *OM:SO₄ was 3.3 (or OC:SO₄ around 4.0). Our ratio of 2.3 ± 3.2 is hence in the same range as previous studies. Murphy et al. [2009] observed that the OC:SO₄ was constant during the first hour of plume dilution into the marine boundary layer. To confirm this at our measurement site, more chemically resolved ship plumes are needed.*

*In addition to a variable chemical content,* there was also a large variation in total mass of the ship
335  plume aerosol,*"*

**2. Specific comments:**

**Referee comment 2.1.**

**P. 1 line 18: Please give percentage fraction of plumes that did not result in measurable secondary**
340  **PM.**

**Author's response 2.1.**

To quantify the fraction of plumes which do and do not produce secondary PM would indeed be valuable information. However, due to the nature of the OFR experiments and our current setup, it is not possible to give an exact fraction. The plumes are smeared in the PAM-OFR and individual
345  secondary PM plumes cannot be distinguished. Hence, during periods with secondary PM formation, we cannot say if all plumes or a fraction of the plumes during such a period contributed to the additional mass, or if it is even ships at all. We have not been able to completely rule out the possibility that it is other sources in the background air which are the main cause of the secondary mass observed, and a longer measurement period would be needed to investigate this further.

350  Still, in order to get a rough idea of the numbers, we made an estimation of the upper and lower fraction of plumes responsible for the secondary PM formation. This is done by assuming that all plumes during hours with secondary PM formation are contributing (upper limit), and then assuming that just one plume per hour with secondary PM formation is contributing (lower limit). During periods without any secondary PM formation, all plumes are counted as not contributing. The calculations are based on the
355  data presented in Figure 3 in the manuscript. A threshold of 1 μg m⁻³ secondary PM mass is used to not include data noise.

This resulted in 39 hours with ship plumes present, but no secondary PM formation, and 19 hours with ship plumes present together with observed secondary PM. So about a third of the times when ship plumes where present, we saw secondary PM formation. During the 19 hours with secondary PM and ship plumes, the total number of ship plumes was 60. We now estimate the upper and lower number of contribution ships, as described above. Assuming only one ship per hour actually causing the secondary PM, results in 19 out of a total of 154 ships, i.e. 12 %. Assuming all ships per hour causing secondary PM results in 60 out of a total of 154 ships, i.e. 39 %. Hence, we conclude that the number of ships which result in measurable secondary PM formation is in the range 10-40 %, and consequently 60-90 % did not contribute to measurable secondary PM.

We have added the upper estimation to the manuscript. And, due to the uncertainties associated with the OFR experiment and the calculations above, we have also highlighted the uncertainties in determining the sources causing the secondary PM observed in order to not cause misunderstandings.

**Author's changes in manuscript 2.1.**

We have added the following text to the manuscript section 3.5 Simulation of atmospheric ageing:

*"Assuming only one plume per hour is actually causing the secondary PM results in 19 out of a total of 154 ships, i.e. 12 %. Assuming all ships sampled in a given hour contributed to the secondary PM results in 60 out of a total of 154 ships, i.e. 39 %. As some of the 154 plumes may individually contribute below the detection limit (indeed this seems likely) and we cannot rule out interferences from non-ship sources, we consider 39% an upper limit on the fraction of ship plumes which contributed to secondary PM in our OFR experiment."*

**Referee comment 2.2.**

**P. 2 line 48 - 50: It would be good to add a review of PM emission factors obtained by the CO2 method, which is probably the most accurate method to infer real-world emissions, and compare to the PM emission factors obtained from testbed experiments.**

**Author's response 2.2.**

See response to comment 1.1.

**Author's changes in manuscript 2.2.**

We have added new text according to the answer to referee comment 1.1.

**Referee comment 2.3.**

**P. 2 line 50 - 51: Please explain: "It is difficult to simulate atmospheric dilution in testbed experiments, which has large effects on nucleated nanoparticles". Please explain the effects on nucleated nanoparticles in more details and add literature references. Does "atmospheric dilution" refer to the fast cooling and expansion of the exhaust plume at stack exit or the dilution due to atmospheric turbulence?**

**Author's response 2.3.**

The intention of this sentence was not mainly to explore details of dilution processes, but to highlight the fact that dilution processes is something that can matter for particle formation, and hence support the need for atmospheric observations in addition to e.g. laboratory studies where conditions might differ. With dilution, we hence mean all relevant processes, including both processes mentioned by the referee, but mainly the first one (fast cooling and expansion at stack exit). Since we do not explore dilution processes further in our manuscript, we find such a review of the literature outside of the main scope. We realize that this sentence and statement might rather cause confusion for the reader. Since the statement does not have a significant impact on our results and our manuscript, we choose to completely remove the sentence.

**Author's changes in manuscript 2.3.**

We have removed this sentence from the manuscript section 1 Introduction:

*It is difficult to simulate atmospheric dilution in testbed experiments, which has large effects on nucleated nanoparticles.*

**Referee comment 2.4.**

**P. 6 line 216 - 220: The explanation of the differences of PM0.15 in ship plumes to the study by Kivekäs et al. (2014) is somewhat speculative. What about the influence of different atmospheric chemistry regime or seasonal differences? The statement "these effects are likely cancelling each other out" is too strong. Simulations and observations of the particle number concentrations in ship plumes reported by Tian et al. (2014, figure 3 therein) show that after rapid dilution in the first minutes, total particle number concentration change only slowly between 30 and 60 minutes. This means that the effect of longer dilution period would be small. On the other hand, the ship contribution determined by Kivekäs et al. covers a larger source corridor, a fact that is not discussed here.**

**Author's response 2.4.**

We agree with the referee and are thankful for pointing out the detailed limitations of the discussion. We have made some statements weaker, in accordance with the uncertainties associated with the results. We have also extended the discussion to include more possible differences between the two measurement sites compared in this paragraph.

**Author's changes in manuscript 2.4.**

The new paragraph in the manuscript section 3.2 Contribution to particle mass concentrations now reads:

"In a similar study to ours by Kivekäs et al. [2014], their reported PM0.15 values were 100 ng m$^{-3}$ within plumes, and 23 ng m$^{-3}$ daily contribution. This compares well to our values of about 50 ng m$^{-3}$ within plumes and 26 ng m$^{-3}$ daily contribution. *However, there are several factors impacting the particle concentrations which differ between the locations, which makes a detailed comparison for e.g. evaluating the effect of SECA regulations difficult. In the study by Kivekäs et al., the data covers a larger ship corridor in the North Sea. Hence,* the ships were *also* larger (had higher gross tonnage and deadweight) *and the particle source is consequently not the same as in Falsterbo. Additionally,* the

distance between the ships and the station is larger in the Kivekäs et al. study, suggesting that plumes *could be* more diluted. *However, the most rapid dilution occurs directly after emission, and then much slower after around 30 minutes of atmospheric transport [Tian et al., 2014], which suggests that this*

435 *should not constitute a large difference between the observed particle concentration between the sites. The particle mass can also be affected by the current chemical state of the atmosphere, e.g. by differences in seasons and meteorology.*

**Referee comment 2.5.**

440 **P. 7 line 267 - 269: The average BC fraction of 2% of the total PM0.5 mass is very low compared to previous studies of ship exhaust, which may be explained in part by different fuels, operating conditions or the use of scrubbers. For a better understanding of this result, please provide the BC fraction, if only the plumes with detectable eBC increase were included in the analysis.**

**Author's response 2.5.**

445 First of all, thanks to drawing our attention to these values, we have found that there is an error in the eBC fractions presented in the results. It now states that the total eBC contribution compared to background levels was 12 % (p. 7, line 251) and that the eBC fraction within ship plumes was 2 % (p. 7, line 254, line 267 and p. 10, line 383). This is incorrect and the values should be 2 % and 5 %, respectively (in agreement with the values presented in table 1).

450 Hence, the eBC fraction is not as low in the ship plumes as previously stated. Still, the eBC fraction is indeed low in ship plumes in Falsterbo in general. The suggestion from the referee to specifically look at the "high BC emitters" is very good and in line with the approach we used to look at the top PM emitters for chemical speciation in Figure 2 in the manuscript. Hence, we selected the 10 % of the plumes ($n = 15$) with the highest eBC mass contributions (all $> 20$ ng m$^{-3}$) and compared these to the

455 corresponding PM$_{0.5}$ contributions, which are calculated from size distributions as described in the manuscript. For these ship plumes, the average $\Delta$PM$_{0.5}$ was ($\pm 1\sigma$) 120$\pm$90 ng m$^{-3}$, and the ship plume eBC fraction was 0.40$\pm$0.20. It should again be remembered that both the eBC and PM$_{0.5}$ plume concentrations have large uncertainties for individual ship plumes due to the low absolute concentrations. Still, there appears to be a substantial fraction of eBC in some plumes, and the total

460 average eBC contribution of 2 % is then consequently a result of many ships having even lower or zero eBC emission. The 10 % of the ship plumes with the highest eBC concentrations were from eleven separate days of the campaign and during different parts of the day, with different meteorological conditions, so there is no apparent bias in the selection of these high eBC-emitters.

**Author's changes in manuscript 2.5.**

465 We have corrected the percentage values of eBC everywhere in the manuscript, so that 2 % refers to the contribution compared to background levels, and that 5 % refers to the plume composition.

We have added the following sentence to the manuscript section 3.4 Contribution the BC and chemical composition:

*"An eBC fraction of 5 % is very low compared to previous ship emission studies. The individual eBC*
470 *fraction varied a lot from ship to ship, which can depend on many factors including engine operation, fuel type, and use of scrubbers. The 10 % of the plumes (n = 15) with the highest eBC mass contribution*

*(> 20 ng m$^{-3}$) had an average eBC fraction (±1σ) of 40±20 % of the PM$_{0.5}$ calculated from SMPS size distributions."*

**Referee comment 2.6.**

475

**P. 9 line 339-340: Please discuss in more detail the (only) moderate increase of nitrate and ammonium in the OFR. Formation of ammonium nitrate requires the presence of sufficient ammonia, which may be the limiting factor during the time when the measurements were made. Thus, nitrate and ammonia could be higher during the spring season with more agricultural**

480

**activity. The gas-phase/particle partitioning is also very sensitive to changes in temperature and relative humidity inside the reactor compared to ambient air.**

**Author's response 2.6.**

The referee is correct that ammonia may have been a limiting factor in particulate nitrate formation. Concentrations of ammonia in Sweden seem to be highest between April-August, up to around 1 µg m$^{-3}$

485

in southern Sweden (1993-2010 average) [*Ferm and Hellsten*, 2012]. Although we did not measure NH$_4$, we expect the concentrations to be lower than the measured NO$_2$ found in table 1 [*Ferm and Hellsten*, 2012]. Using the OH exposures of the reactor and a reaction rate of 9.2*10$^{-12}$ cm$^3$ molecules$^{-1}$ s$^{-1}$ [*Mollner et al.*, 2010] between 50 and 90 % (depending on the OH concentrations which in turn depends mostly on the absolute humidity) of the NO$_2$ entering the reactor should have

490

reacted. This translates to between 2-10 µg m$^{-3}$ of nitrate. However, to form particulate nitrate the formed HNO$_3$ must be neutralized by the limited ammonia but is also limited by the reactor condensation sink, in a similar fashion as for LVOCs (Fig S4). Setting up a model for nitrate is possible, but not necessary for any conclusions in this manuscript. The temperature effect in reactors could be significant and is known to affect both organics and ammonium nitrate [*Nenes et al.*, 2020]. However,

495

if it was important in this campaign, the nitrate already in the particle phase should also have evaporated, and the difference between reactor and ambient measurements would be negative, which was not the case. Further, pH of the particles, in combination with liquid water content affects the sensitivity to ammonia and nitrate. A lot of the above discussion is important for ambient OFR studies, but does not affect any of the conclusions in the present manuscript.

**Author's changes in manuscript 2.6.**

500

We have added the following to the manuscript section 3.5 Simulation of atmospheric processing:

"Modelling of the fate of produced low-volatile species (Supporting Information and Fig. S4) suggests that a significant portion (~60-90%) of the oxidation products do not enter the particle phase due to the low condensation sink. *Although the model was set up for organics, this is true for all secondary aerosol*

505

*species formed in the reactor. Particulate nitrate formation will also be sensitive to the availability of gas-phase ammonia to neutralize the aerosol."*

**Referee comment 2.7.**

**P. 9 line 351-353: How likely is it that heterogeneous oxidation of organic aerosols happens in the real plume ageing where the environment is much less oxidative?**

**Author's response 2.7.**

Heterogeneous oxidation surely happens in the atmosphere as well, but is likely different in an OFR, since the particle surfaces in the atmosphere are not as static as in the OFR, due the dynamics of a specific air mass (emissions and mixing). *Renbaum and Smith* [2011] showed that the radical concentration and time are interchangeable variables in lab experiments, i.e. OFR experiments can simulate the atmospheric oxidation, but that heterogeneous oxidation can be affected by adsorption of $O_3$ to particle surfaces. We don't find it necessary to add any of this discussion to the manuscript.

**Author's changes in manuscript 2.7.**

No changes were done to the manuscript.

**3. Technical corrections:**

**Referee comment 3.1.**

**P. 4 line 143: "Chemical composition" would be better here.**

**Author's response 3.1.**

We agree with the referee.

**Author's changes in manuscript 3.1.**

We have changed the word "*content*" to "*composition*".

**Referee comment 3.2.**

**Figure 1: Please add a panel with wind direction and wind speed below the current plots.**

**Author's response 3.2.**

We think that the suggestion from the referee to include wind data, is mainly in order to show that the changes in particle concentration are not due to these factors but indeed due to ships as we state in the manuscript. As can be seen in the Fig. AC3 below, the wind speed and direction are fairly constant during the period (255-282 degrees and 7.6-9.6 m s$^{-1}$). We do not think that these figures are contributing with enough relevant information to be included in the manuscript. However, we recognize the potential risk of other readers wondering about the same thing, hence we have added a comment about this in the figure caption.

[Figure]

Figure AC4. Wind direction and wind speed during the same period which is shown in the Fig. 1 in the manuscript.

**Author's changes in manuscript 3.3.**

We have added the following phrase to the end of the caption of Figure 1:

"*The wind direction (269±14 degrees) and wind speed (8.7±1.1 m s⁻¹) can be considered stable during this period.*"

**Referee comment 3.3.**

**Figure 3: The orange line for ammonium is hardly visible.**

**Author's response 3.3.**

We appreciate the feedback on the visibility of the data in the plot. We agree that both ammonium and nitrate are not clearly visible, but this is mainly due to the often zero change in concentration of these species. In order to improve the middle panel of Figure 3, we made a new version with the relatively brighter line for ammonium in front for comparison (see Figure AC5 and AC6 below). The figures are similar, but due to the colors of the lines (which are the standard colors for these aerosol species within the field), we think that the new version (AC6) is slightly more clear.

[Figure]

Figure AC5.

[Figure]

Figure AC6.

555

**Author's changes in manuscript 3.3.**

560     We have replaced Figure 3 (now Figure 4) with the new version, where all the data is identical but the ammonium concentration time series is moved to the front (i.e. the Figure AC6 above).

**Referee comment 3.4.**

**Table 1: Please give the number of plumes in summer and winter somewhere in this table. It**
565     **would be better to replace "average plume concentration" by "Delta plume" since the column gives the excess concentration due to the ship plume and not the in-plume concentration.**

**Author's response 3.4.**

We agree with the referee regarding the use of "delta plume" instead of "average plume concentration". Regarding the number plumes, this is already given in the table. The rightmost column shows the
570     number of plumes ($n$) per season and per aerosol variable. To make this even clearer, we have added an explanation in the table headline.

**Author's changes in manuscript 3.4.**

We have changed the table heading in Table 1 from "*Average plume conc.*" To "$\Delta plume$".

We have added the phrase "*a number (n) of*" to the table headline in Table 1, which now reads:

575     "*Table 1: Average absolute contribution of particle mass (PM), NO$_2$, equivalent black carbon (eBC), particle number concentration (PN), SO$_2$, and CO$_2$ to local air quality due to a number (n) of ship plumes, from two measurement campaigns at the Falsterbo coastal site.*"

**Final author comment for Referee 2**

580    ## 1. Major/general comments:

**Referee comment 1.1.**

**The evaluated impact on of ship plume is really low, by reading the manuscript I do not understand what are the authors conclusions, is this due to the application of SECA rules? Or to other meteorological effect? This aspect is fundamental to give guideline to environmental**

585    **policies. Please authors improve the discussion on this aspect.**

**Author's response 1.1.**

We acknowledge that the conclusions regarding the cause of the low PM is not clear. However, we also think that with the current data set, we cannot exclusively point to a simple explanation for the observed levels. Since there are no similar measurements in the region before the SECA, we cannot compare the

590    PM levels before and after regulation. We could find several explanations for the observed low PM contribution from ships in this study. Firstly, the SECA regulation could indeed be a cause of PM reduction, as is now more clearly stated in the introduction (see Author's response 1.1 to referee 1). Secondly, we only consider one shipping lane with the method used, and the total contribution from shipping is then larger due to ship emissions being a source to the regional background aerosol. We

595    raise this point in the description of the method, in the discussion, and in the conclusions, specifically on p. 3-4, line 121-124, on p. 6, line 230, on p. 7, line 275, and on p. 10, line 388-390 in the first version of the manuscript and have clarified this also in the abstract and in the introduction (see Author's response 1.3 to referee 1) Thirdly, the ships in Falsterbo are smaller than in the rest of the Baltic and on larger seas elsewhere. This could be an additional reason for a lower impact at our field site. And, since

600    most other ambient studies have been performed on ship fleets with larger sized vessels, this will affect comparisons with emission factors from literature. We do not think the observations of low PM are due to meteorology only, even if meteorology has an impact on observations (see Author's response 1.5 to referee 1). This is partly based on the fact that measurements were performed in both summer and winter and yielded similar results. We have extended the discussion to make all of the previously not mentioned

605    factors above more clear to the reader.

**Author's changes in manuscript 1.1.**

We have made the following changes to the manuscript section 3.2 Contribution to particle mass concentrations:

"Most ships have a small contribution to PM, of less than 100 ng m$^{-3}$. *A relatively low PM is expected*

610    *in the strictest SECA regions compared to elsewhere, due to the strong reduction in particulate sulphate. However, there are multiple reasons for the low PM contribution observed in Falsterbo. The fuel sulphur content and the small, but existing, use of scrubbers is the first explanation, and the compliance with these regulations is indeed high. Secondly, the impact presented is from one shipping lane, and the total contribution from shipping is hence larger due to ship emissions being a source to*

615    *the regional background aerosol. Thirdly, the ships in Falsterbo are relatively small compared to the rest of the Baltic and on larger seas elsewhere, since they have to pass the Øresund Strait and under bridges. Smaller ships typically have a lower engine power and emit less air pollutants on an absolute*

*scale. Meteorological factors did not seem to influence the plume detection to any large extent, and the shipping lane contribution to ambient concentrations was similar in both the summer and winter campaign. However, longer measurements would be needed to study the effect of meteorological parameters on plume detection, and this will be of even larger importance in other measurement sites where there is risk of a build-up of a thermal internal boundary layer at the shoreline."*

**Referee comment 1.2.**

**The result of the scarce increases of secondary aerosol obtained in the OFR measurements is not fully supported. I think that a comparison with data from other sites could help the discussion (e.g. Contini et al., 2011 and especially Perez et al., 2016 for the evaluation of secondary aerosol).**

**Author's response 1.2.**

We do not fully understand what the referee means with "not fully supported". The OFR results show increases in secondary aerosol production in some occasions, while not in others. It is important to understand the limitations of the OFR measurement technique, but since this is a minor part of the manuscript, we do not want to extend the description of these results too much. See also the Author's response 2.1 to referee 1, about an upper estimation of ships contributing to secondary aerosol.

Regarding the references suggested, we think these are not easily comparable to our results. Contini et al. (2011) describes diurnal patterns of primary $PM_{2.5}$ and $PM_{10}$, not measurement or modelling of secondary aerosol formation. There is a statement in the introduction about the possible contribution to secondary inorganic aerosol due to sulphur, and a statement about the difficulties to extract the contribution to secondary aerosol from the data. In Perez et al. (2016), the SOA is obtained from PMF and they explicitly state the difficulties in quantification and highlight the need for further research. We have chosen not to add any detailed comparison with the suggested papers, but rather our manuscript highlights the first ambient OFR studies on ship emissions and the associated observations.

**Author's changes in manuscript 1.2.**

We have not made any changes to the manuscript.

**Referee comment 1.3.**

**Besides, the statement that background particles are already aged is not correct in my opinion, as the author state the time required to air masses from ship plume to the sampling site is 90 min in the measuring condition, this time is not sufficient for the ageing of aerosol, nether in summer.**

**Author's response 1.3.**

Here, we think there is a misunderstanding about what is meant with "background particles". With "air masses being somewhat aged" we mean air that is not coming from the shipping lane, but the regional background. This aerosol is hence much more aged than the stated 90 minutes, more like days up to a week. That this air is at least partly aged is shown in the ambient AMS O:C ratio in figure 4 [*Canagaratna et al.*, 2015]. We have made some linguistic changes in the manuscript in order to make this unambiguous.

**Author's changes in manuscript 1.3.**

We have revised the following sentence in the manuscript section 3.5 Simulation of atmospheric processing:

"This may be caused by the *background* air masses reaching the site already being somewhat aged and precursor concentrations being low."

And revised the following sentences in the manuscript section 4 Summary and conclusions:

"We suggest that the reason for this is that the *regional* background particles *from long-range transport* arriving at Falsterbo are already relatively aged."

**Referee comment 1.4.**

**I think that information on the height of planetary boundary layer are fundamental to understand secondary aerosol formation processes and especially the real contribution of ship aerosol to background aerosol budget. By considering this aspect, conclusions can change substantially. I strongly suggest adding a discussion on the effect of PBL height on the contribution of both primary and secondary ship aerosol respect to background.**

**Author's response 1.4.**

The topic of planetary boundary layer effects was also brought up by referee 1. We have therefore treated these comments together, see the response to referee 1 ("Referee comment 1.5").

Regarding secondary aerosol formation, we are not sure how the referee means that this should be affected in a different way compared to the primary aerosol. As stated in the manuscript, the PAM-OFR measurements were limited in time and we cannot make any conclusive statements regarding this matter.

**Author's changes in manuscript 1.4.**

See the response to referee 1 (Referee comment 1.5).

**2. Minor comments:**

**Referee comment 2.1.**

**A map with the sampling site and surrounding areas with the urbanization level can be useful to interpret the data set.**

**Author's response 2.1.**

We agree with the referee. Additionally, since both referees brought up boundary layer effects as a concern, the map will also help the reader to interpret the geographical and meteorological conditions (see Author's response 1.5 to referee 1).

**Author's changes in manuscript 2.1.**

We have added the following figure and caption to the manuscript section 2 Materials and methods. The map is now "Figure 1" in the manuscript, and all other figures have changed number accordingly.

[Figure]

*"Figure 1. Map of the Baltic Sea region (left) and the nearby region around the field site (right). The measurement site Falsterbo is marked with blue pin, and the nearby large cities Copenhagen and Malmö are marked with red circles. The red line shows a typical route of a ship following the main shipping lane around the Falsterbo peninsula, based on AIS position data."*

Additionally, the following text has been updated in section 2 Materials and methods:

"The field site and the measurement methods have been described in Ausmeel et al. [2019] and is only briefly outlined here. The measurements took place at the Falsterbo peninsula, Southern Sweden, during January-March and May-July, 2016. *The location of the field site and the surrounding area are shown in Fig. 1. The largest nearby cities are Copenhagen (Capital of Denmark) and Malmö, with populations of about 800 000 and 300 000, respectively. All water in this figure is within the North Sea and Baltic Sea SECA.*  *The field site is located at the tip of a peninsula, around which a frequently trafficked shipping lane is passing (illustrated by a red line in Fig. 1)."*

**Referee comment 2.2.**

**Section 3.1 Plume identification and general characteristic seems more a methodology to recognize ship plume than results, I suggest moving this part in the Materials and methods section.**

**Author's response 2.2.**

We think that just a part of the section could be considered to be method, while a large part of the section is relevant to keep in the results, since it describes the general observations of plumes and evaluation of successful measurement techniques. We have moved a part of this section, in order to keep methods and results more clearly separated.

**Author's changes in manuscript 2.2.**

The following text has been moved from section '3.1 Plume identification and general characteristics' to section '2 Materials and methods'. In order to make the text fit into the new location, a few structural and linguistic changes have been made, this is the new text:

"The average concentration for each plume was calculated by integrating the total area under the plume peak. The values were then normalized by the plume duration to give an average plume peak concentration. All ship passages that resulted in an elevated particle number concentration, *fulfilling the criteria for plume selection listed above,* and which could be connected to an individual ship with AIS were included in the calculation of the average contribution from the fleet. Daily and seasonal *contribution* values *from the shipping lane* are *calculated* based on AIS data, which showed an average of 73 and 63 ships passing per day in winter and summer respectively. During periods when the wind blows from the Øresund Strait (i.e. across the shipping lane), the Falsterbo site is *hence* affected by the nearest shipping lane approximately 51% of the time in the winter, and 44% in the summer, based on the average observed plume duration of 10 min *multiplied with the average number of ships per day*. Based on historical wind data from the last 20 years (Swedish Meteorological and Hydrological Institute, SMHI), the wind intercepts the shipping lanes in Øresund Strait about 70% of the time in both summer and winter, which was used to estimate the seasonal contribution from ships. For the daily and seasonal estimates, it was assumed that the average *ship plume* contribution *(Δplume)* in Table 1 is representative for all plumes. For calculation of the uncertainty in the daily and seasonal contribution, the uncertainty in aerosol number concentration was estimated to 30 %, the uncertainty in particle loss estimation was 30 %, the variation in ship traffic density was 17-34 %, and the uncertainty in seasonal wind pattern was estimated to 5 %. These values were used to calculate the total uncertainty with error propagation, i.e. added in quadrature."

**Referee comment 2.3.**

**Lines 339-340. The sentence is not completely correct, it is true that nitrate arises from oxidation of NOx, but ammonium arises from neutralization of ammonia on both HNO3 and H2SO4, the latter is preferred over ammonium nitrate formation due to the lower vapor pressure of sulfuric acid than nitric acid (Hauglustaine et al., 2014).**

**Author's response 2.3.**

We have removed the incomplete explanation of the chemical origin of the (low) formation of particulate nitrate and ammonium and revised the sentence.

**Author's changes in manuscript 2.3.**

The revised sentence reads:

*The increases in nitrate and ammonium were moderate on an absolute scale.*

65   One way to characterize and quantify ship emissions is through ambient measurements in coastal areas, downwind of shipping lanes. This method makes it possible to register an increase in aerosol levels and potential exposure in an area when individual ship emission plumes pass the measurement station. Other methods include e.g. measurements on laboratory engine emissions [*Anderson et al.*, 2015; *Kasper et al.*, 2007; *Lyyränen et al.*, 1999; *Petzold et al.*, 2010] or measurements on board or following a sailing ship, intersecting the emission plume [*Chen et al.*, 2005; *Murphy et al.*, 2009; *Petzold et al.*, 2008; *Aliabadi et al.*, 2016; *Lack et al.*,

70   2011]. However, while these methods can provide detailed knowledge on fresh emissions from a specific ship, they do not give information about the variety of particle properties between different ships, how the plume evolves during transport in the atmosphere, human exposure over land areas, and these methods can be more cost-intensive. By measuring in ambient conditions on the coast, emissions from a large part of the shipping fleet can be captured, and atmospheric measurements are needed to give information on emissions, dilution, and impact on environment and local air quality. Atmospheric measurements of elevated CO$_2$

75   concentrations close to (less than a few minutes downwind) shipping lanes, can give emission factors during atmospheric conditions, which differ from testbed conditions.

To date, a number of atmospheric studies of individual ship plumes have been conducted in harbour areas [*Alföldy et al.*, 2013; *Healy et al.*, 2009; *Jonsson et al.*, 2011; *Lu et al.*, 2006; *Westerlund et al.*, 2015], and also in the Arctic [*Aliabadi et al.*, 2015]. One

80   study of aged plumes from a shipping lane has been performed by *Kivekäs et al.* [2014], outside the west coast of Denmark, measuring plumes with an atmospheric age of about one hour. In the study by Kivekäs et al., ship plumes were measured at a coastal station about 25-50 km from the shipping lane, where there was good potential to study the impact of ship emissions on land

concentrations and how particles are aged during semi-long range transport. However, the authors suggested shorter distances to be able to detect elevated particle number concentrations from each individual ship passing along the lane when winds blew from the ships to the coastal stations. In this study, we have performed measurements 10 km (corresponding to approximately 30 minutes of plume aging) downwind of a major shipping lane in southern Sweden. With this setup, we were able to measure elevated particle number concentration for a majority of the ship plumes. The distance is nevertheless large enough to represent typical shipping lanes around the globe which influence inland air-pollution, as well as to observe some effects of particle aging.

The measurements presented here were performed in the Baltic Sea SECA during 2016 in order to study ship emission properties after the newest regulation of fuel sulphur content (0.1% by mass). In a report by *Mellqvist et al.* [2017], the compliance levels to the most recent SECA regulations was studied in the nearby region of where our measurements were conducted. During the years 2015 and 2016, the compliance level was 92 %–94 % in the region around Denmark. Also *Jonson et al.* [2019] have shown that there is a strong indication that ships are complying, based on emission modelling before and after the 2015 SECA regulation. The method for individual ship plume identification and the contribution to particle number concentrations have been described in detail in *Ausmeel et al.* [2019]. In the current paper however, we report the contribution from a major shipping lane to local particle mass concentrations and chemical composition (organics, sulphate, black carbon), and NO$_2$, as well as the effects of additional aging simulated with an oxidation flow reactor. The results complement previous studies in two ways. Firstly, due to the new measurement location at an intermediate distance from the shipping lane. Secondly, due to the measurements being performed after the recent fuel sulphur regulations within SECAs, which was introduced on January 1, 2015. The estimation of how ship traffic along a major route contributes to the coastal particle concentrations can contribute to the development of aerosol dynamic process models, regional aerosol particle models, health assessment models and epidemiological studies.

**2 Materials and methods**

The field site and the measurement methods have been described in Ausmeel et al. [2019] and is only briefly outlined here. The measurements took place at the Falsterbo peninsula, Southern Sweden, during January-March and May-July, 2016. The location of the field site and the surrounding area are shown in Fig. 1. The largest nearby cities are Copenhagen (Capital of Denmark) and Malmö, with populations of about 800 000 and 300 000, respectively. All water in this figure is within the North Sea and Baltic Sea SECA.  The field site is located at the tip of a peninsula, around which a frequently trafficked shipping lane is passing (illustrated by a red line in Fig. 1). When performing measurements along a coastline, the interface between atmospheric boundary layers over land and sea can cause a build-up of a thermal internal boundary layer at the shoreline. This has been described by e.g. *Hanna et al.* [1984] and is important to consider when doing plume measurements. However, due to the geographical surroundings at our field site, we do not expect this effect to be strong or exist at all, since the peninsula is a small land mass almost surrounded by water. The boundary layer height is likely a parameter which affects the dilution of the plumes and the background air and thereby the plume concentration. However, it seemed that boundary layer height did not affect the plume concentration in a systematic and reproducible way in Falsterbo as deduced using Hybrid Single Particle Lagrangian Integrated Trajectory Model (HYSPLIT) [*Stein et al.*, 2016] with meteorological data from Global Data Assimilation System (GDAS). Since this study aims at describing the ship emission contribution to inland aerosol burden and not the concentrations at the exhaust stack, the effects of boundary layer height has not been pursued any further in this paper.

[revised manuscript text omitted]

Individual ship plumes were extracted from the data set based on a set of criteria, which are described in the companion paper [*Ausmeel et al.*, 2019]. In brief, plumes were initially selected by inspection of the time series, choosing peaks in particle number concentration where there was a clear increase above the background and noise level. This increase should not be longer than about 20 minutes and not shorter than about 5 minutes, to exclude other potential sources than ship plumes from the nearby lane. In order to assess individual ship plumes, no plumes that were overlapping or close to each other in time (less than five minutes) were included in the analysis. Plumes were only selected when the wind was blowing over the shipping lane to the measurement station. Further, automatic ship identification system (AIS) position data was used to confirm that the increase in particles was due to a passing ship. This was performed by calculating a trajectory of the emission plume from the ship using wind data. The expected plume passage based on AIS and wind data and the visible plumes in the particle counter agreed excellently, and a large fraction (>95 %) of the plumes (including all plume events, not only the plumes finally used for analysis for which 100 % were confirmed with AIS) were detectable as a distinct increase compared to background particle concentrations. Background concentrations were subtracted from the total plume concentration to get only ship emission contributions. The background level in this analysis was defined as the baseline concentration from which the identified plumes can be distinguished (the background concentration was

always relatively stable during onshore wind, see Fig. 2, and hence the background subtraction was straightforward). 
[revised manuscript text omitted]
 or in the Baltic Sea in general. However, according to *Jonson et al.* [2015], the fuel distribution in the Baltic and North Sea in 2011 was around 74 % residual (e.g. HFO) and 26 % was distillate fuels (MDO, MGO). Further, *Johansson and Jalkanen* [2016] reported a 88 % decrease in SO$_x$ and 36 % decrease in PM$_{2.5}$ emissions from the year 2014 to 2015 in the Baltic Sea, based on AIS data and emission modelling. Strong decreases in SO$_x$ and PM$_{2.5}$ due to SECA implementations have also been predicted and reported in e.g. *Kalli et al.* [2013] and *Jonson et al.* [2015]. Regarding the use scrubbers to reduce airborne sulphur emissions, in the paper by Jonson et al. [2019] it is stated that there was an increase in the number of scrubbers used globally, from 77 vessels in 2014 to 155 in 2016. Out of these, 85 vessels with scrubbers were operating in the Baltic Sea area in 2016. This is a small fraction of the > 20 000 ships operating in the region [*Johansson and Jalkanen*, 2016].~~Another potential explanation for the variation in plume properties could be meteorological factors. In this study we have considered wind speed and precipitation, but no detailed analysis of the plume dispersion was performed and is outside the scope of this paper. Large variations between individual ships was also shown by *Jonsson et al.* [2011] at a measurement distance of 0.5-1 km from the ships.~~

[revised manuscript text omitted]

Cooper, D. A. (2001), Exhaust emissions from high speed passenger ferries, Atmos. Environ., 35, 4189-4200, https://doi.org/10.1016/S1352-2310(01)00192-3.

Corbett, J. J., and Fischbeck, P. (1997), Emissions from Ships, Science, 278, 823-824, 10.1126/science.278.5339.823.

Corbett, J. J., Winebrake, J. J., Green, E. H., Kasibhatla, P., Eyring, V., and Lauer, A. (2007), Mortality from Ship Emissions: A Global Assessment, Environ. Sci. Technol., 41, 8512-8518, 10.1021/es071686z.

Corbin, J. C., Pieber, S. M., Czech, H., Zanatta, M., Massabò, D., Orasche, J., El Haddad, I., Mensah, A. A., Stengel, B., Drinovec, L., Mocnik, G., Zimmermann, R., Prévôt, A. S. H., and Gysel, M. (2018), Brown and Black Carbon Emitted by a Marine Engine Operated on Heavy Fuel Oil and Distillate Fuels: Optical Properties, Size Distributions, and Emission Factors, J. Geophys. Res. Atmos., 123, 6175-6195, 10.1029/2017jd027818.

Corbin, J. C., Czech, H., Massabò, D., de Mongeot, F. B., Jakobi, G., Liu, F., Lobo, P., Mennucci, C., Mensah, A. A., Orasche, J., Pieber, S. M., Prévôt, A. S. H., Stengel, B., Tay, L. L., Zanatta, M., Zimmermann, R., El Haddad, I., and Gysel, M. (2019), Infrared-absorbing carbonaceous tar can dominate light absorption by marine-engine exhaust, npj Climate and Atmospheric Science, 2, 12, 10.1038/s41612-019-0069-5.

Diesch, J.-M., Drewnick, F., Klimach, T., and Borrmann, S. (2013), Investigation of gaseous and particulate emissions from various marine vessel types measured on the banks of the Elbe in Northern Germany, Atmos. Chem. Phys., 13, 3603-3618.

Drinovec, L., Močnik, G., Zotter, P., Prévôt, A., Ruckstuhl, C., Coz, E., Rupakheti, M., Sciare, J., Müller, T., and Wiedensohler, A. (2015), The" dual-spot" Aethalometer: an improved measurement of aerosol black carbon with real-time loading compensation, Atmos. Meas. Tech., 8, 1965-1979.

Eyring, V., Köhler, H. W., van Aardenne, J., and Lauer, A. (2005), Emissions from international shipping: 1. The last 50 years, J. Geophys. Res. Atmos., 110, 10.1029/2004jd005619.

Eyring, V., Isaksen, I. S. A., Berntsen, T., Collins, W. J., Corbett, J. J., Endresen, O., Grainger, R. G., Moldanova, J., Schlager, H., and Stevenson, D. S. (2010), Transport impacts on atmosphere and climate: Shipping, Atmos. Environ., 44, 4735-4771, https://doi.org/10.1016/j.atmosenv.2009.04.059.

Hanna, S. R., Paine, R. J., and Schulman, L. L. (1984), Overwater dispersion in coastal regions, Boundary-Layer Meteorology, 30, 389-411, 10.1007/BF00121963.

Healy, R. M., O'Connor, I. P., Hellebust, S., Allanic, A., Sodeau, J. R., and Wenger, J. C. (2009), Characterisation of single particles from in-port ship emissions, Atmos. Environ., 43, 6408-6414.

Huang, C., Hu, Q., Wang, H., Qiao, L., Jing, S. a., Wang, H., Zhou, M., Zhu, S., Ma, Y., Lou, S., Li, L., Tao, S., Li, Y., and Lou, D. (2018), Emission factors of particulate and gaseous compounds from a large cargo vessel operated under real-world conditions, Environ. Pollut., 242, 667-674, https://doi.org/10.1016/j.envpol.2018.07.036.

IPCC (2013), Climate Change 2013: The Physical Science Basis. Contribution of Working Group I to the Fifth Assessment Report of the Intergovernmental Panel on Climate Change, Cambridge University Press, Cambridge, United Kingdom and New York, NY, USA, 1535 pp., www.climatechange2013.org, ISBN 978-1-107-66182-0.

Johansson, L., and Jalkanen, J.-P. (2016), Emissions from Baltic Sea shipping in 2015, HELCOM Baltic Sea Environment Fact Sheets. Online. 2020-05-20, http://www.helcom.fi/baltic-sea-trends/environment-fact-sheets/.

Jonson, J. E., Jalkanen, J. P., Johansson, L., Gauss, M., and Denier van der Gon, H. A. C. (2015), Model calculations of the effects of present and future emissions of air pollutants from shipping in the Baltic Sea and the North Sea, Atmos. Chem. Phys., 15, 783-798, 10.5194/acp-15-783-2015.

Jonson, J. E., Gauss, M., Jalkanen, J.-P., and Johansson, L. (2019), Effects of strengthening the Baltic Sea ECA regulations, Atmos. Chem. Phys., 19, 13469-13487.

Jonsson, Å. M., Westerlund, J., and Hallquist, M. (2011), Size-resolved particle emission factors for individual ships, Geophys. Res. Lett., 38, 10.1029/2011gl047672.

Kalli, J., Jalkanen, J.-P., Johansson, L., and Repka, S. (2013), Atmospheric emissions of European SECA shipping: long-term projections, WMU Journal of Maritime Affairs, 12, 129-145, 10.1007/s13437-013-0050-9.

[revised manuscript text omitted]

Petzold, A., Lauer, P., Fritsche, U., Hasselbach, J., Lichtenstern, M., Schlager, H., and Fleischer, F. (2011), Operation of Marine Diesel Engines on Biogenic Fuels: Modification of Emissions and Resulting Climate Effects, Environ. Sci. Technol., 45, 10394-10400, 10.1021/es2021439.

Sandradewi, J., Prévôt, A. S. H., Szidat, S., Perron, N., Alfarra, M. R., Lanz, V. A., Weingartner, E., and Baltensperger, U. (2008), Using Aerosol Light Absorption Measurements for the Quantitative Determination of Wood Burning and Traffic Emission Contributions to Particulate Matter, Environ. Sci. Technol., 42, 3316-3323, 10.1021/es702253m.

Shen, F., and Li, X. (2019), Effects of fuel types and fuel sulfur content on the characteristics of particulate emissions in marine low-speed diesel engine, Environmental Science and Pollution Research, 10.1007/s11356-019-07168-6.

Sinha, P., Hobbs, P. V., Yokelson, R. J., Christian, T. J., Kirchstetter, T. W., and Bruintjes, R. (2003), Emissions of trace gases and particles from two ships in the southern Atlantic Ocean, Atmos. Environ., 37, 2139-2148, https://doi.org/10.1016/S1352-2310(03)00080-3.

Stein, A. F., Draxler, R. R., Rolph, G. D., Stunder, B. J. B., Cohen, M. D., and Ngan, F. (2016), NOAA's HYSPLIT Atmospheric Transport and Dispersion Modeling System, Bulletin of the American Meteorological Society, 96, 2059-2077, 10.1175/bams-d-14-00110.1.

Svenningsson, B., Arneth, A., Hayward, S., Holst, T., Massling, A., Swietlicki, E., Hirsikko, A., Junninen, H., Riipinen, I., and Vana, M. (2008), Aerosol particle formation events and analysis of high growth rates observed above a subarctic wetland–forest mosaic, Tellus B: Chemical and Physical Meteorology, 60, 353-364.

Tian, J., Riemer, N., West, M., Pfaffenberger, L., Schlager, H., and Petzold, A. (2014), Modeling the evolution of aerosol particles in a ship plume using PartMC-MOSAIC, Atmos. Chem. Phys., 14, 5327-5347, 10.5194/acp-14-5327-2014.

[revised manuscript text omitted]